# The Paf1 complex positively regulates enhancer activity in mouse embryonic stem cells

Li Ding[1], Maciej Paszkowski-Rogacz[1] ⓘ, Jovan Mircetic[1,2], Debojyoti Chakraborty[1], Frank Buchholz[1,3,4] ⓘ

The RNA polymerase II (RNAPII) associated factor 1 complex (Paf1C) plays critical roles in modulating the release of paused RNAPII into productive elongation. However, regulation of Paf1C-mediated promoter-proximal pausing is complex and context dependent. In fact, in cancer cell lines, opposing models of Paf1Cs' role in RNAPII pause-release control have been proposed. Here, we show that the Paf1C positively regulates enhancer activity in mouse embryonic stem cells. In particular, our analyses reveal extensive Paf1C occupancy and function at super enhancers. Importantly, Paf1C occupancy correlates with the strength of enhancer activity, improving the predictive power to classify enhancers in genomic sequences. Depletion of Paf1C attenuates the expression of genes regulated by targeted enhancers and affects RNAPII Ser2 phosphorylation at the binding sites, suggesting that Paf1C-mediated positive regulation of pluripotency enhancers is crucial to maintain mouse embryonic stem cell self-renewal.

## Introduction

Transcription of many eukaryotic protein-coding genes is regulated in large part by enhancers, DNA sequences that increase the likelihood that transcription of a particular gene will occur under favorable conditions (1). Enhancers are found in intergenic regions, introns and exons, and can activate transcription independently of their location, distance, or orientation with respect to promoters (2). Enhancers play a central role in spatiotemporally orchestrating gene expression programs, and alterations of enhancer activities are frequently implicated in diseases. Therefore, the identification and molecular characterization of enhancers is an important research field. A range of methods have been developed to predict enhancers, based on their characteristics including transcription factor binding, chromatin accessibility, histone modifications, or promoter–enhancer interactions. However, none of these methods correlate perfectly with enhancer activity because most active enhancers carry only partial characteristic marks (summarized in reference 3). In contrast to the indirect prediction methods, a recently developed technique named self-transcribing active regulatory region sequencing (STARR-seq) allows direct survey of active enhancers by coupling enhancer activity to its sequence in *cis*. STARR-seq has been applied genome-wide in flies and in some mammalian cells and has greatly advanced our understanding of how enhancer activities are encoded in the genome (4, 5).

Promoters and enhancers share common features such as an open chromatin structure and an accumulation of transcription factor binding sites (6, 7). Recent studies have revealed another common feature, showing that both genetic elements are transcribed by RNAPII, leading to the production of mRNAs and enhancer RNAs (eRNAs), respectively (8, 9, 10). A critical step leading to active transcription of mRNA is RNAPII release from pausing at promoters, whereby a highly regulated chain of events allow RNAPII to proceed to transcriptional elongation (11, 12). Widespread control of transcriptional pausing and elongation has also been reported at enhancers, where RNAPII is retained at enhancers and eRNA synthesis requires pause-release. This finding suggests that RNAPII pausing control at enhancers might be a critical layer of enhancer regulation (13).

The RNA polymerase-associated factor 1 complex (Paf1C) has been identified as a crucial checkpoint for RNAPII promoter-proximal pausing and pause-release at both protein-coding genes and at enhancers. However, divergent effects during promoter-proximal pausing and pause-release of Paf1C have been reported in different cancer cell lines (14, 15, 16). In one study, Paf1C was reported to maintain promoter-proximal pausing of RNAPII in a colon cancer cell line by inhibiting the positive transcription elongation factor b (14), whereas in another study, Paf1C was shown to promote release from pausing in a lymphoma cell line (15). The latter study proposed a positive feedback model in which Paf1C promotes pause-release by regulating the chromatin association of positive transcription elongation factor b (17). Hence, the Paf1C-mediated regulation of promoter-proximal pausing in higher eukaryotes appears to be complex and context dependent (18). In agreement with the negative

[1]Medical Systems Biology, Medical Faculty and University Hospital Carl Gustav Carus, Technische Universität Dresden, Dresden, Germany   [2]Mildred Scheel Early Career Center, National Center for Tumor Diseases Dresden (NCT/UCC), Medical Faculty and University Hospital Carl Gustav Carus, Technische Universität Dresden, Dresden, Germany   [3]National Center for Tumor Diseases (NCT), Medical Faculty and University Hospital Carl Gustav Carus, Technische Universität Dresden, Dresden, Germany   [4]German Cancer Research Center (DKFZ), Heidelberg and German Cancer Consortium (DKTK) Partner Site, Dresden, Germany

Correspondence: frank.buchholz@tu-dresden.de
Debojyoti Chakraborty's present address is Council of Scientific and Industrial Research Institute of Genomics and Integrative Biology, New Delhi, India

role of Paf1C in promoter-proximal pausing and pause-release in the colon cancer cell line, Chen and colleagues also reported that the Paf1C represses a subset of enhancers by restricting eRNA transcription in the same cell line (14, 16).

Our group has previously identified Paf1C as an important factor to maintain mouse embryonic stem cells (mESCs) pluripotency by specifically regulating the expression of pluripotency genes (19, 20). Paf1C binds to promoters of pluripotency genes, where it is required to maintain a transcriptionally active chromatin structure. Depletion of Paf1C leads to a decreased expression of these pluripotency genes, accompanied by a loss of mESC self-renewal and subsequent differentiation. Thus, Paf1C may impact on the expression of pluripotency genes by maintaining the chromatin structure accessibility of RNAPII, or regulating the release of promoter-proximally paused RNAPII for active transcription in mESCs. To expand our understanding of Paf1Cs' role in promoter and enhancer regulation during pluripotency, we performed a panel of experiments to characterize its molecular function.

# Results

### Ctr9 exhibits cell type–specific chromatin binding patterns

To elucidate molecular functions of Paf1C, we generated a GFP-tagged knock-in mESC cell line, where GFP is fused to the 3′ terminus of Ctr9, a core component of the Paf1C. To compare results obtained in mESCs, we also generated a Ctr9-GFP knock-in line in mouse NIH3T3 cells (Fig S1). These cell lines were used to perform chromatin immunoprecipitation coupled with DNA sequencing (ChIP-seq, single runs) using a validated anti-GFP antibody (21, 22) to investigate the genome-wide binding pattern of Ctr9 in these cell types. Analyses of the ChIP-seq experiments identified 33,412 and 13,471 cell type–specific Ctr9 binding sites in mESCs and NIH3T3 cells, respectively, whereas 5,673 common binding sites were identified in both cell types (Table S1). 76% of the Ctr9 binding sites were found at protein-coding genes, whereas 24% of the binding sites mapped to intergenic region in mESCs (Fig S2A). Similarly, 78% and 22% of the Ctr9 peaks mapped to protein-coding genes and intergenic regions in NIH3T3 cells (Fig S2B). Consistent with its role as a regulator of transcription elongation, Ctr9 was highly enriched at transcription start sites (TSSs) and gene body of protein-coding genes in both cell lines (23, 24). In addition, we observed intensive Ctr9 occupancy around transcript end sites, which have been proposed to regulate alternative cleavage and polyadenylation (APA) in mammalian cells (Fig 1A and B) (25). To investigate whether Ctr9 binding correlates with active transcription, we performed mRNA sequencing (RNA-seq) in mESCs and NIH3T3 cells, and quantified Ctr9 binding intensity for genes of different expression levels. Markedly, Ctr9 binding positively correlated with mRNA level in both cell types, where strong Ctr9 peaks were observed on genes of high expression (Fig 1A and B). Of note, however, not all strongly expressed genes were bound by Ctr9 (Fig S2C and D), suggesting that Paf1C binding is not essential for high-level transcription of genes per se. Consistent with its role as an elongation factor associated with active gene transcription, Ctr9 binding exhibited cell

type–specific binding patterns at protein-coding genes. For example, Ctr9 was specifically enriched at most known pluripotency genes including *Oct4*, *Sox2*, *Nanog*, *Esrrb*, and *Zfp42* in mESCs (Figs 1C and S2E and F), whereas a panel of fibroblast marker genes (e.g., *Col1a1*, *Col1a2*, and *Col5a1*) were only bound by Ctr9 in NIH3T3 cells (Figs 1C and S2G and H).

### Occupancy of Ctr9 and NELF on protein-coding genes

Several recent studies have reported Paf1C as a key factor that regulates promoter-proximal pausing (14, 15, 26). However, possible roles that Paf1C might play to release RNAPII from the poised to the active state are still controversial. To determine how Paf1C regulates promoter-proximal pausing in relation to NELF (a four-subunit protein complex that negatively impacts transcription elongation by RNAPII) in mESCs, we GFP-tagged the NELF subunit NELFA (Fig S3) (21). Chromatin occupancy of NELFA and RNAPII Ser5p were determined by ChIP-seq using an anti-GFP antibody and an anti-RNAPII Ser5p antibody in NELFA-tagged mESCs. Results were then compared with the ChIP-seq results from the Ctr9 pull-down. We first inspected NELFA, RNAPII Ser5p and Ctr9 manually at several transcribed genes. Interestingly, for most of these genes, NELFA, RNAPII Ser5p and Ctr9 peaks did not perfectly overlap. Whereas the NELFA and RNAPII Ser5p peaks localized around the TSS, Ctr9 peaks summited slightly downstream. For instance, the peaks of NELFA and RNAPII Ser5p at the *Actb* gene summited around the TSS, whereas little Ctr9 binding was observed at this region. However, along with the decrease in NELFA and RNAPII Ser5p, Ctr9 reads accumulated progressively (Fig 1D). This suggests that Paf1C, NELFA and RNAPII Ser5p do not co-occupy the same sequence at the TSS of protein-coding genes. Indeed, a metagene profile analysis confirmed that NELFA was specifically enriched close to TSS regions (21 bps downstream of TSSs on average), whereas Ctr9 was barely detectable at this position (Fig 1E and F and Table S2), arguing against the proposal that Paf1C and NELFA co-bind to RNAPII to establish promoter-proximal pausing (14, 18).

Inspection of RNAPII Ser5p peaks revealed that they accumulated downstream of NELFA (39 bps downstream TSSs), suggesting that after initiation, RNAPII is paused at the proximal promoters by NELF. With the decrease in NELFA and RNAPII Ser5p, Ctr9 piled up and reached its maximum level around 145 bps downstream of the RNAPII Ser5p peaks (184 bps downstream of the TSS) (Fig 1E and F). These data suggest that the switch from transcription initiation (with NELFA and RNAPII Ser5p occupancies) to transcription elongation (with Paf1C occupancy) happen around +1 nucleosome (146 bps) downstream of the RNAPII promoter-proximal pausing sites, consistent with a previous report showing that the general elongation complex is established within a narrow window of 150 bps downstream of the TSS (27).

### Ctr9 is enriched at super enhancers in mESCs and marks active enhancers

The genome-wide ChIP-seq analysis revealed that ~24% of the binding sites for Ctr9 were found in intergenic regions (Fig S2A and B). To investigate possible roles of Paf1C at these positions, we compared the binding sites with previously published ChIP-Seq

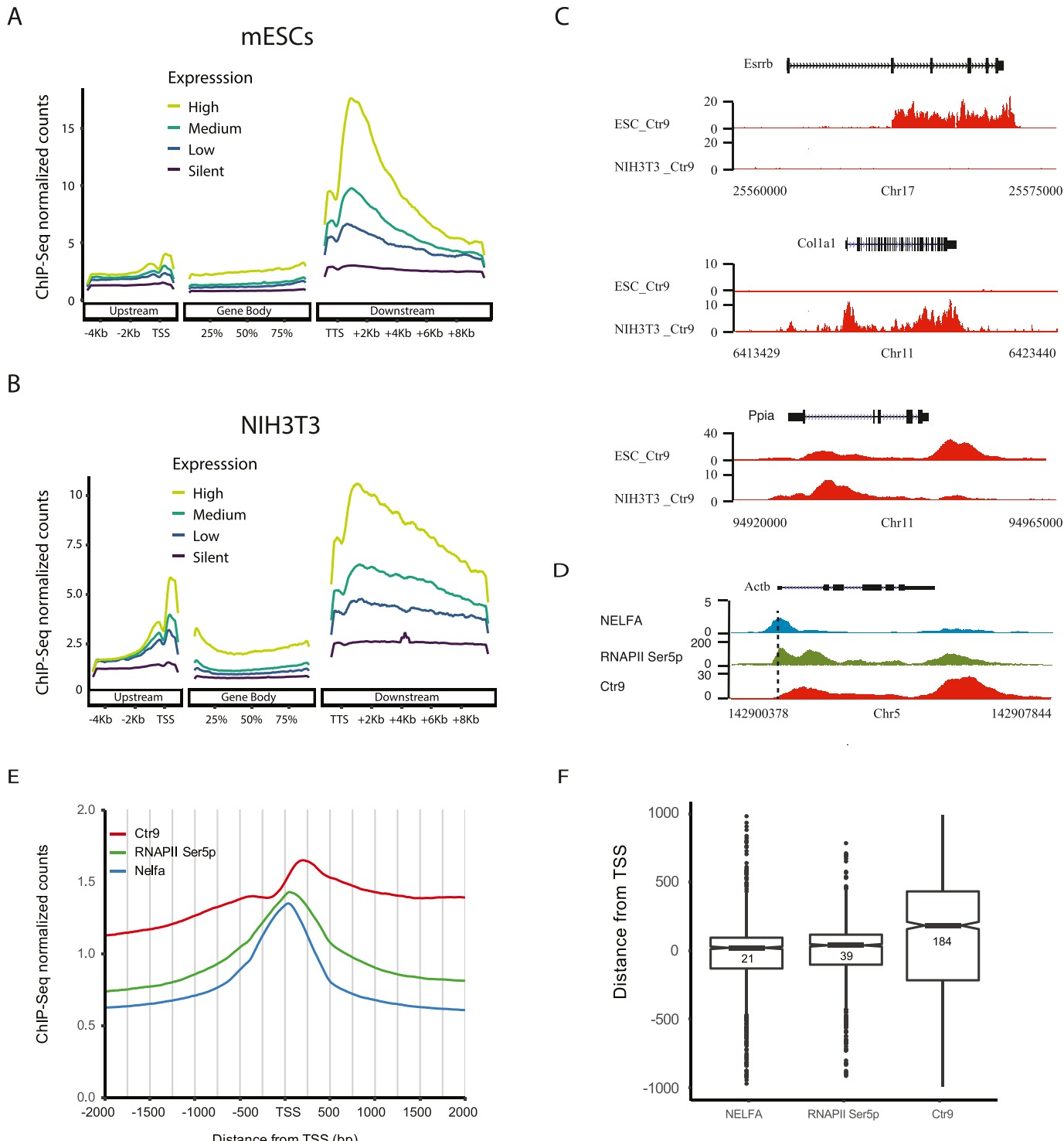

**Figure 1. NELF, RNAPII Ser5p, and Paf1C occupancy at protein-coding genes.**
**(A, B)** Metagene analyses showing positive correlation between Ctr9 occupancy and gene expression levels in mouse embryonic stem cells (A) and NIH3T3 (B). Based on expression data (RNA-seq), genes were divided into high (top 25%), medium (25–75%), low (bottom 25%), and silent gene expression categories. Ctr9 binding intensities to the upstream, gene body, and downstream part of the gene of each category were calibrated. Each ChIP-Seq experiment was performed with a single sample. **(C)** Representative genome browser tracks of Ctr9 ChIP-seq in mouse embryonic stem cells (upper) and NIH3T3 cells (lower). The x-axis indicates the chromosome position, and the y-axis represents normalized read density in reads per million. *Esrrb* (Top panel) and *Col1a1* (middle) are shown for ESC and NIH3T3 cell specific binding, respectively, whereas *Ppia* (bottom) is shown as example for Ctr9 binding to both cell types. **(D)** NELFA (blue), RNAPII Ser5p (green), and Ctr9 (red) occupancy at the *Actb* gene. The x-axis indicates the chromosome position, and the y-axis represents normalized read density in reads per million. Note the shift of the Ctr9 peak with respect to the NELFA and RNAPII Ser5p peaks. The dotted black line marks the transcription start site (TSS) of the *Actb* gene. **(E)** Metagene profiles of ChIP-seq read coverages

data for various transcription factors and chromatin modifications. Strikingly, we observed a strong overlap of the Ctr9 binding sites with binding sites for H3K27ac, and to a lesser extend also to H3K4me1 (Table S3), two histone modifications that have been shown to mark enhancers in mammalian cells (28). The overlap was particularly strong at so called super enhancers (SEs), a small portion of enhancers with unusually strong enrichment for binding of transcriptional coactivators (29, 30). For instance, Ctr9 peaks overlapped with H3K27ac and H3K4me1, and were enriched on the annotated *Oct4* enhancers (Fig 2A) (31). Hence, similar to cancer cell lines (16), Paf1C binds to enhancer sequences in mESCs. To gain a systematic overview of Paf1C binding to enhancers, we analyzed Ctr9 occupancy on the 231 described SEs and on 8,563 typical enhancers (TEs) in mESCs (29). 72% SEs and 7% TEs were bound by Ctr9, suggesting that Ctr9 binding was more prominent on SEs than on TEs (Table S3) (29, 30). Furthermore, Ctr9 occupancy density on SEs was significantly higher (2.7 folds higher, $P < 2.2 \times 10^{-16}$) than on TEs (Fig 2B).

SEs associate with many biological processes, which define cell identity, and exhibit strong cell type specificity (30). We therefore decided to inspect Ctr9 binding at intergenic regions in NIH3T3 cells, and compared the data with that in mESCs. Indeed, our analysis revealed substantial differences of Ctr9 binding between mESCs and NIH3T3 cells. Consistent with a specific enhancer function only in mESC, 131 SEs (57%) were uniquely bound by Ctr9 in these cells, with many of these SEs found in the vicinity of well-known pluripotency genes, including *Oct4*, *Nanog*, *Sox2*, *Esrrb*, and *Tbx3*. In contrast, only 35 of the mESC SEs (15%) were also bound by Ctr9 in NIH3T3 cells (Table S4). Many of these common Ctr9-bound SEs were found in the vicinity of genes regulating fundamental biological processes, such as metabolism (*Dusp1*, *Uck2*, *Elovl6*, and *Sgk1*), transportation (*Slc6a6* and *Ssr2*), scaffold (*Mesd* and *Spry2*), stress response (*Nfkbia* and *Gadd45a*), and RNA processing (*Gbx2* and *Gtf3c6*), suggesting that these SEs serve a similar function in mESCs and NIH3T3 cells.

H3K4me1 is thought to mark active enhancers, and more prominently poised enhancers, whereas H3K27ac is believed to mark active enhancers only (28). To determine how Paf1C regulates enhancers in relation to histone modifications, we analyzed the overlap of Ctr9 with H3K27ac, and H3K4me1 on the annotated SEs and TEs (Fig 2C and D) (32). Both H3K4me1 and H3K27ac marks were highly enriched on SEs, with 90% and 88% of the SEs carrying the modifications, respectively. In contrast, 83% and 31% of the TEs were found to carry the H3K4me1 and H3K27ac modifications respectively, suggesting that most TEs are poised enhancers (H3K4me1+/H3K27ac−). Interestingly, almost all Ctr9-bound SEs and TEs carried H3K27ac modification, suggesting that Ctr9 predominantly associates with active enhancers (Fig 2C and D and Table S3).

To functionally investigate the two categories of SEs, those with Ctr9 binding (SE/Ctr9+) and those without Ctr9 binding (SE/Ctr9−), we cloned 21 randomly selected SEs with Ctr9 binding and 28 SEs without Ctr9 binding into a plasmid that reports enhancer activity when transfected into mammalian cells. Importantly, this assay has

previously demonstrated to faithfully reflect in vivo enhancer activity (33, 34). The Dual-Luciferase reporter assays revealed that although all tested SEs possessed H3K27ac marks, the ones without Ctr9 binding (SE/Ctr9−) were inactive or far less active than those with Ctr9 binding (SE/Ctr9+) ($P < 4.89 \times 10^{-7}$), suggesting that more active enhancers are bound by Ctr9 (Fig 2G and Table S5). Thus, Ctr9 occupancy may help to nominate the most active enhancers from poised/inactive enhancers, more reliably than H3K27ac marks alone.

### Ctr9 binding correlates with RNAPII Ser5p, Ser2p, and NELFA at SEs

The Ctr9 occupancy at active enhancers encouraged us to check whether other components of the transcription apparatus associate with Paf1C to regulate enhancer activities on these sites. We first analyzed NELFA enrichment at SEs. 119 NELFA binding sites were detected for the 166 SEs (71%) that showed Ctr9 binding, suggesting that most active SEs are regulated by both Paf1C and NELF complexes (Figs 3A and S5A and Table S4). The distance between NELFA and Ctr9 peaks on SEs (465 bps) were significantly larger than that on the protein-coding genes (163 bps), suggesting that enhancer (e)RNA transcription may be orchestrated in a different way than protein-coding genes (Fig 3B). Interestingly, on most SEs, NELFA and Ctr9 binding exhibited unilateral patterns, suggesting that eRNAs transcription predominantly proceeds into one direction (1D eRNAs; Fig 3C) (8). However, for a few SEs, we observed that one NELFA peak was flanked by two Ctr9 peaks, indicating that eRNA transcription may proceed to both directions as proposed for 2D eRNAs at these enhancers (9) (Fig 3D).

Paf1C has been shown to associate with RNAPII Ser5p and Ser2p at promoters and the coding regions of genes (35). However, a systematic investigation of RNAPII occupancy in relation to Paf1C at enhancers is still pending. To fill this gap, we performed RNAPII Ser2P and Ser5p ChIP-seq experiments using antibodies against RNAPII Ser2p and Ser5p in mESCs. Consistent with previous reports, our analysis revealed that RNAPII Ser5p is mainly detected at the TSS regions, whereas RNAPII Ser2p is primarily found at gene bodies and exhibited maximum occupancy around transcription termination sites of protein-coding genes, reflecting the Ctr9 occupancy pattern (Fig S4) (36). In addition to protein-coding genes, we found that RNAPII Ser5p and Ser2p were also significantly enriched on SEs, with 83% and 63% occupancy, respectively, in line with the notion that SEs are transcribed by RNAPII (Fig 3A and Table S4). Strikingly, almost all Ctr9-bound SEs were also positive for RNAPII Ser5p and all 146 Ser2P-bound SEs were co-occupied by Ctr9. Moreover, RNAPII Ser2P was hardly detectable on SEs devoid of Ctr9 bindings, suggesting that the Paf1C cooperates with RNAPII phosphorylation to regulate the activity of SEs in mESCs (Figs 3A, 4A–C, and S5B and Table S4).

Bulk accumulation of Ctr9 and RNAPII Ser2p at enhancers that correlated with high enhancer activities prompted us to inspect in situ enhancer transcription. We aligned our Ctr9 and RNAPII Ser2p ChIP-seq data with a published global run-on sequencing (GRO-seq)

---

across 4-kb windows centered around the TSSs of all genes bound by Ctr9. The y-axis shows an average normalized read count scaled to 10 million reads. **(F)** Box plot of the binding positions of NELFA, RNAPII Ser5p and Ctr9 around the TSS region with peaks detected in ChIP-seq experiments. The y-axis shows distances in base pairs of peaks to the annotated TSS (calculated with the software Homer) (53).

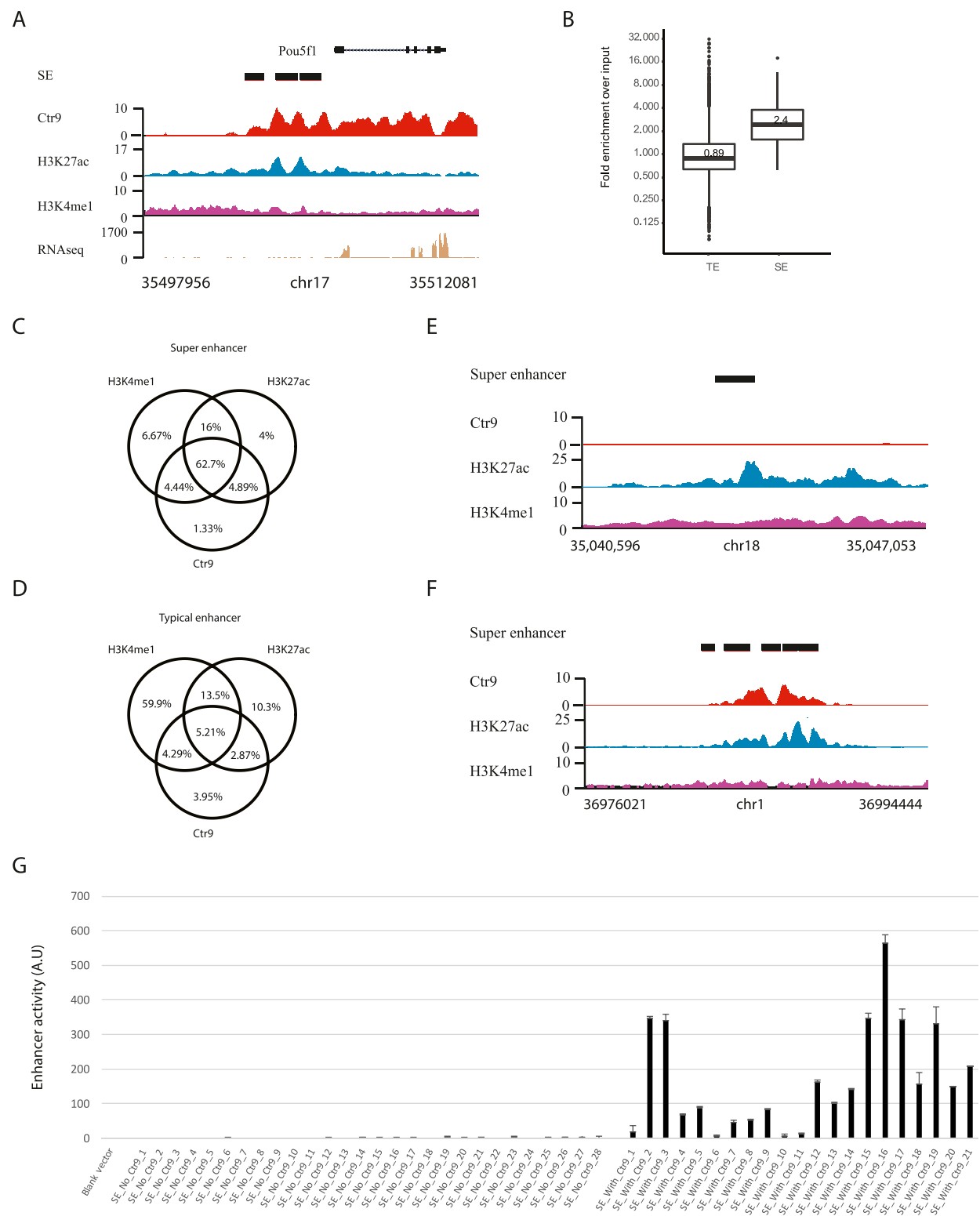

**Figure 2. Paf1C is enriched at enhancers, and correlates with enhancer activities.**
**(A)** Genome browser tracks of ChIP-seq results obtained for Ctr9, H3K27ac, and H3K4me1, and RNA-seq results in the vicinity of the *Pou5f1* (*Oct*4) gene in mES cells. The x-axis indicates the chromosome position, and the y-axis represents normalized read density in reads per million. Black boxes indicate annotated *Pou5f1* enhancers. Each ChIP-Seq experiment was performed with a single sample. **(B)** Box plots of Ctr9 binding densities on typical enhancers (TEs) and super enhancers (SEs). Significantly higher levels of Ctr9 binding was measured comparing SEs with TEs. *P*-value < 2.2 × 10$^{-16}$ according to Wilcoxon rank sum test (SEs versus TEs). **(C)** Venn diagram analysis of H3K27ac, H3K4me1, Ctr9 occupancy on SEs. The numbers represent the percentages of SEs with corresponding histone modifications, and/or Ctr9 binding. **(D)** Venn

data set, which can be applied to detect eRNA transcripts in mESCs (12). Analysis revealed remarkable overlap between Ctr9, RNAPII Ser2p and eRNAs transcripts (Figs 4A, B, and D and Table S4). A total of 126 SEs expressed eRNA transcripts, all of which were bound by Ctr9, and almost all (124) were also occupied by RNAPII Ser2p, suggesting that Paf1C and RNAPII Ser2p occupancy is a hallmark for active eRNAs transcription in mESCs.

### Paf1C regulates enhancer activity and RNAPII Ser2p

To investigate whether Paf1C directly regulates enhancer activity in mESCs, we performed Ctr9 knockdown experiments (Fig S1C). 24 h post transfection, 10 luciferase reporter plasmids harboring SE fragments with Ctr9 binding (SEs/Ctr9+), and 10 SE fragments without Ctr9 binding (SEs/Ctr9−) were transfected into Ctr9 knockdown cells. As control, the same reporter plasmids were also transfected into cells treated with a non-targeting silencing trigger. Dual-Luciferase assays revealed that depletion of Ctr9 significantly reduced the enhancer activity of SEs/ Ctr9+ (four enhancers exhibits more than twofold decrease in activities), whereas no significant effects were recorded on SEs/Ctr9−. These results suggest that Ctr9 is required to maintain enhancer activity in mESCs (Fig 5A and B and Table S6). To determine if Ctr9 regulates transcription of target gene through the modulation of enhancer activity, we depleted Ctr9 by RNAi in mESCs followed by RNA-seq analysis. Ctr9 depletion resulted in a marked decrease in expression of SE target genes (73.5%), whereas TE target genes were much less effected (63.8%) and genes not associated with enhancers were typically not effected (53.7%) (Fig 5C and D and Table S7).

It has been reported that depletion of the Paf1C leads to reduction in phosphorylation of RNAPII CTD Ser2 on protein-coding genes, which is mediated by Paf1C-dependent recruitment of the CTD Ser2 kinase CDK12 to RNAPII (15, 37). To determine if a similar mechanism might regulate the RNAPII Ser2p status on SEs, we depleted Ctr9 by RNAi in mESCs, followed by ChIP-seq against RNAPII Ser2p. Ctr9 depletion resulted in a 1.5-fold decrease of RNAPII Ser2P occupancy over SEs and a 1.3-fold depletion over TEs, suggesting that Paf1C regulates RNAPII Ser2p on SEs in a similar way as on protein-coding genes (Fig 5E and Table S4). Thus, SEs with a strong decrease of RNAPII Ser2p after Ctr9 knockdown lost their enhancer activity dramatically, and this effect correlated with a strong decrease in transcription of their associated target genes, such as *Oct4* and *Tbx3* (Figs 5F–I and S6A–D). In contrast, Ctr9 depletion had no significant effects on RNAPII Ser2p and mRNA transcripts for genes without Ctr9 binding on SEs (e.g., *Zfp638*, Fig S6E–H). Together, our analyses suggest that Paf1C regulates gene expression by modulating enhancer activity.

### Prediction of active enhancers in mESCs and NIH3T3 cells

Although histone modification H3K27ac has been shown to mark active enhancers, our data suggest that H3K27ac in conjunction with

Ctr9 binding improves the prediction to pinpoint the most active enhancers. To investigate whether a combination of H3K27ac and Ctr9 binding can predict strong enhancers in cell types where enhancers have not yet been fully characterized, we analyzed H3K27ac and Ctr9 binding in NIH3T3 cells, and nominated 4,037 active enhancers (Table S8) in this cell type. To test our prediction experimentally, we selected eight sites of H3K27ac+/Ctr9− and four sites of H3K27ac+/Ctr9+ in both mESCs and NIH3T3 cells, and eight sites of H3K27ac+/Ctr9+ only in NIH3T3 cells. The enhancer activity was determined again using the luciferase reporter system in the respective cell lines. All four sites of H3K27ac+/Ctr9+ in mESCs and NIH3T3 cells showed marked enhancer activity in both cell lines, whereas seven out of eight H3K27ac+/Ctr9+ NIH3T3-specific regions exhibited prominent enhancer activities only in NIH3T3 cells. Furthermore, all eight H3K27ac+/Ctr9 regions were inactive in both cell types (Fig 6A and B and Table S9). Hence, our data suggest that active enhancers in specific cell types can be reliably predicted based on H3K27ac and Paf1C occupancy.

# Discussion

At promoters, the Paf1C has been described to be a major regulator of promoter-proximal pausing and transcriptional elongation, but in different cancer cell lines, contradictory models describing the mechanism of Paf1C function in this respect have been proposed (14, 15, 26). To investigate the mechanism of Pac1C regulating promoter function in primary stem cells, we have performed a panel of experiments to determine DNA occupancies of relevant factors, including NELFA, RNAPII Ser5p, RNAPII Ser2P, and Ctr9 in mESCs. Our analyses are consistent with a dynamic transition of NELF, RNAPII Ser5p, and Paf1C within a 150-bp region downstream of the TSS (27), indicating a coordinated action of events of these factors for active transcription in mESCs, where NELF is substituted by Paf1C. Hence, our model for Paf1Cs role in promoter-proximal pausing and transcriptional elongation at promoters is more in line with the data described by Yu et al (15). Notably, our data are also in agreement with a recent cryo-EM structure analysis of activated and paused RNAPII elongation complexes showing that Paf1C and NELF bind to RNAPII in a mutually exclusive fashion, so that transcription elongation is facilitated by the release of NELF and recruitment of Paf1C to RNAPII (38).

In addition to occupancy at protein-coding genes, we observed prominent Ctr9 binding to intergenic regions, which frequently overlapped with previously described enhancer markers. Furthermore, our data revealed a positive correlation between Ctr9 binding and the strength of activity of enhancers. Notably, ChIP-seq data unmasked strong enrichment of Ctr9 on most SEs in mESCs. What is more, functional testing of Paf1C-bound SEs confirmed that

---

diagram analysis of H3K27ac, H3K4me1, Ctr9 occupancy on TEs. The numbers represent the percentages of TEs with corresponding histone modifications, and/or Ctr9 binding. **(E, F)** Representative genome browser tracks for Ctr9, H3K27ac, and H3K4me1 at SEs without Ctr9 binding (E), or with Ctr9 binding (F). The x-axis indicates the chromosome position, and the y-axis represents normalized read density in reads per million. Black boxes indicate the annotated SEs. **(G)** Experimental evaluation of SE activities. Indicated DNA elements were tested to drive expression of the firefly luciferase gene. The y-axis shows the luciferase measurements in A.U. The values are normalized to samples transfected with the empty reporter vector. The pRL-SV40 plasmid was used as transfection efficiency control. Data are presented as the mean ± SD from three independent experiments. Error bars depict SD.

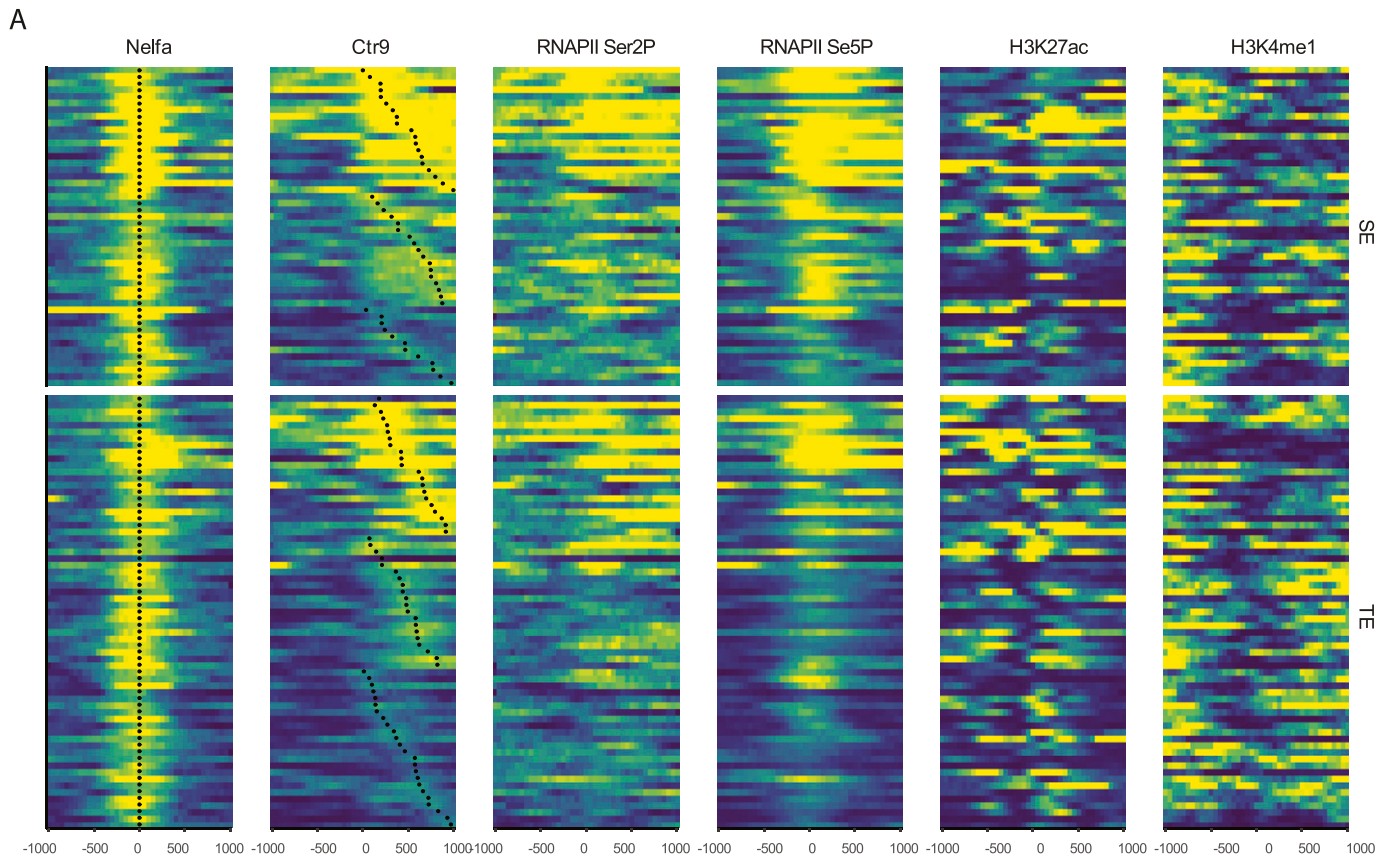

**A**

Nelfa    Ctr9    RNAPII Ser2P    RNAPII Se5P    H3K27ac    H3K4me1

SE

TE

-1000 -500 0 500 1000 -1000 -500 0 500 1000 -1000 -500 0 500 1000 -1000 -500 0 500 1000 -1000 -500 0 500 1000 -1000 -500 0 500 1000

Distance from summits of Nelfa ChIP-Seq peaks, oriented towards Ctr9 ChIP-Seq peak summits

Signal quantile (per sample)

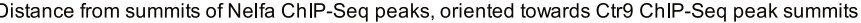

0.00 0.25 0.50 0.75

**B**

Distance between Nelfa and Ctr9 binding sites

TEs    354

SEs    465

**C**

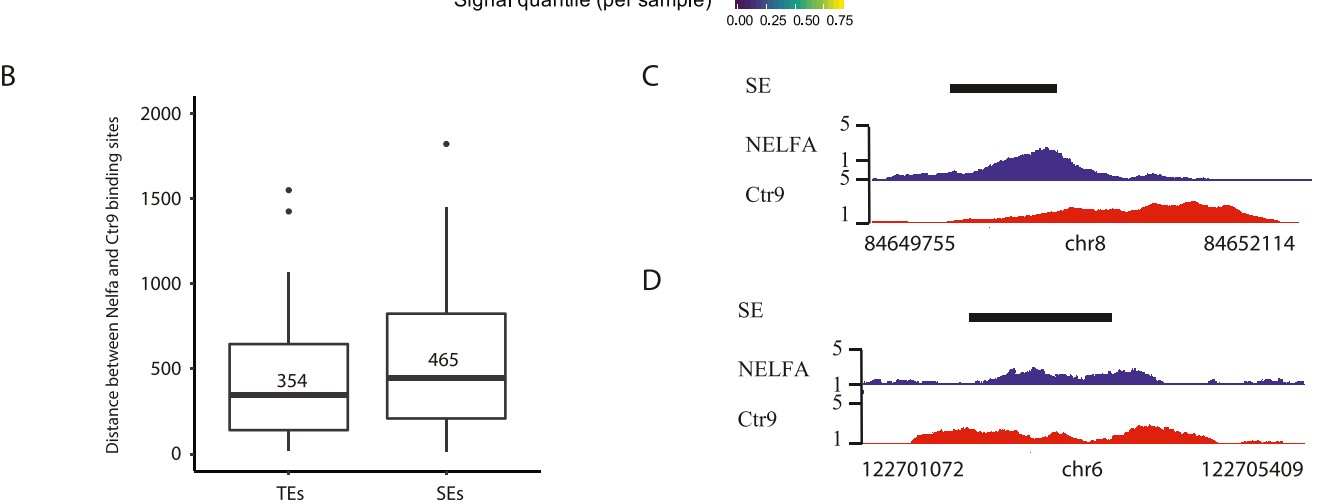

SE

NELFA

Ctr9

84649755    chr8    84652114

**D**

SE

NELFA

Ctr9

122701072    chr6    122705409

**Figure 3.   Occupancy of NELFA and Ctr9 at enhancers.**
**(A)** Heat maps showing ChIP-Seq occupancy patterns of NELFA, Ctr9, RNAPII Ser2P, RNAPII Ser5P, H3K27ac, and H3K4me1 at enhancers that are bound by NELFA. Color-coding is based on quantile-normalized read coverage signals, with yellow indicating stronger binding. Black dots covering NELFA and Ctr9 heat maps are placed at summits of predicted ChIP-Seq peaks. All heat maps are showing 2-kb windows centered around NELFA peak summits, oriented towards the nearest Ctr9 peak summit. Each ChIP-Seq experiment was performed with a single sample. **(B)** Box plots showing distance in base pairs between NELFA and Ctr9 binding sites on typical enhancers and super enhancers (SEs). The average numbers in base pairs for the two classes are presented within the plots. Distances of peaks were measured between predicted peak summits, as reported by the peak calling software Homer (53). **(C, D)** Representative genome browser tracks for Ctr9 and NELFA ChIP-seq at SEs showing unilateral (C) or bilateral (D) Ctr9 binding on SEs in relation to NELFA. The x-axis indicates the chromosome position, and the y-axis represents normalized read density in reads per million. Black boxes indicate annotated SEs.

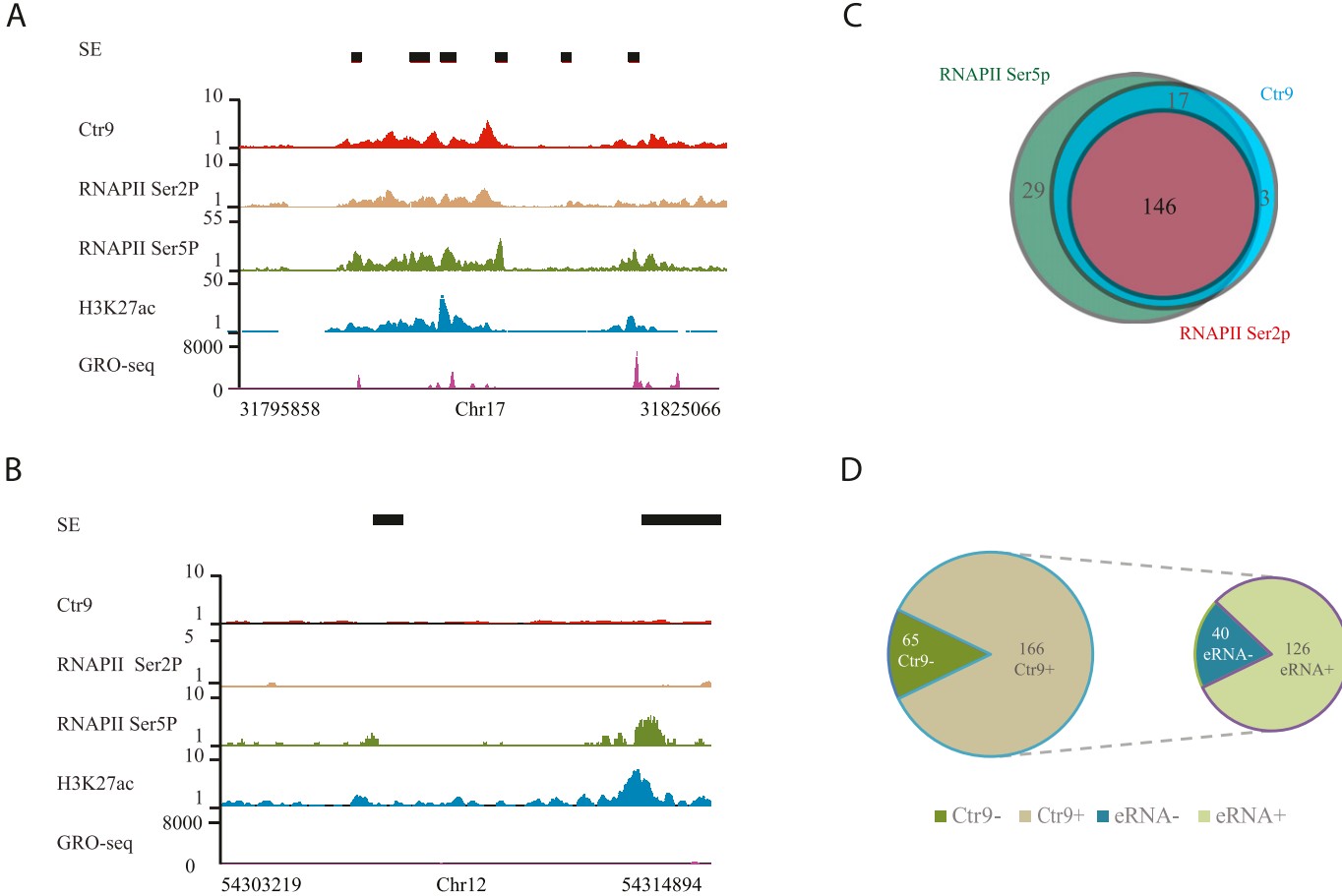

**Figure 4. Paf1C and RNAPII Ser2p correlate with enhancer RNA (eRNA) transcripts.**
**(A, B)** Representative genome browser tracks showing RNAPII Ser2p, RNAPII Ser5p, H3K27ac and eRNA on super enhancers (SEs) with Ctr9 binding (A) or without Ctr9 binding (B). **(A, B)** RNAPII Ser5p and H3K27ac are detectable in both (A) and (B), whereas RNAPII Ser2p and eRNA transcripts are detected only on SE with Ctr9 binding (A). The x-axis indicates the chromosome position, and the y-axis represents normalized read density in reads per million. Black boxes indicate annotated SEs. Each ChIP-Seq experiment was performed with a single sample. **(C)** Venn diagram analysis of RNAPII Ser2p, RNAPII Ser5p, and Ctr9 occupancy on SEs. The numbers indicate SEs carrying either histone modifications or Ctr9 binding. **(D)** Analysis of Ctr9 occupancy and eRNA transcripts on SEs. The numbers indicate SEs with Ctr9 binding and eRNA transcripts. The plot shows that all 126 eRNAs were transcribed from SEs with Ctr9 binding.

these sequences acted as strong enhancers in mESCs. In contrast, a subset of SEs void of Ctr9 binding were not able to drive expression in reporter assays. Hence, enhancer activity highly correlated with Ctr9 binding, indicating that Ctr9 binding marks the most active enhancers in mESCs. In addition, depletion of Ctr9 strongly reduced activities of these SEs concomitantly with a decrease in RNAPII Ser2p occupancy, supporting a model in which Paf1C binding stimulates eRNA transcription at enhancers (Fig 6C). Because eRNAs have been shown to play an important role in the strength of enhancer activity, we propose that Paf1C is implicated in eRNA transcription at enhancers, thus contributing to their activity (39, 40, 41, 42).

Our findings are divergent to a recent study, which reported occupancy of Paf1C at enhancers in a human colon cancer cell line to repress enhancer activity and to curb eRNA transcription (16). Data by these authors are consistent with a model in which Paf1C maintains the paused state of RNAPII at enhancers and promoters. Thus, regulation of enhancers by Paf1C may be cell type–specific, as it appears to be in the case of Paf1C regulation of promoter-proximal pausing (15, 16, 18). Such difference may be especially pronounced between cancer cells and

primary cells and more work is required to investigate potential alternations of Paf1Cs' role in RNAPII pause-release in different cell types.

Finally, we show that Paf1C binding can be used in combination with histone marks to predict active enhancers in mESCs and NIH3T3 cells. RNAPII binding and eRNA transcription have been reported to displace nucleosomes and establish DNA accessibility (43, 44). Therefore, active enhancers are transcribed even if this transcription has no effects *in trans*. Based on eRNA transcription, pipelines have been designed to independently predict regulatory elements in the genome without using chromatin marks (7, 45). Although eRNAs are potent indicators of enhancer activity, eRNA-based enhancer prediction could be distorted by important factors, including stability of eRNA, or the sensitivity of eRNA detection. Consequently, Andersson et al reported that up to 33 percent of tested regions with enhancer activity lacked detectable eRNAs (9). Similarly, we observed that 24% of the SEs with Ctr9 binding did not express detectable eRNAs, whereas enhancer activity could clearly be measured. Our data suggest that the binding of Paf1C marks enhancers more reliably and robustly than eRNA transcription or H3K27ac marks alone, and therefore, could serve as a better predictor

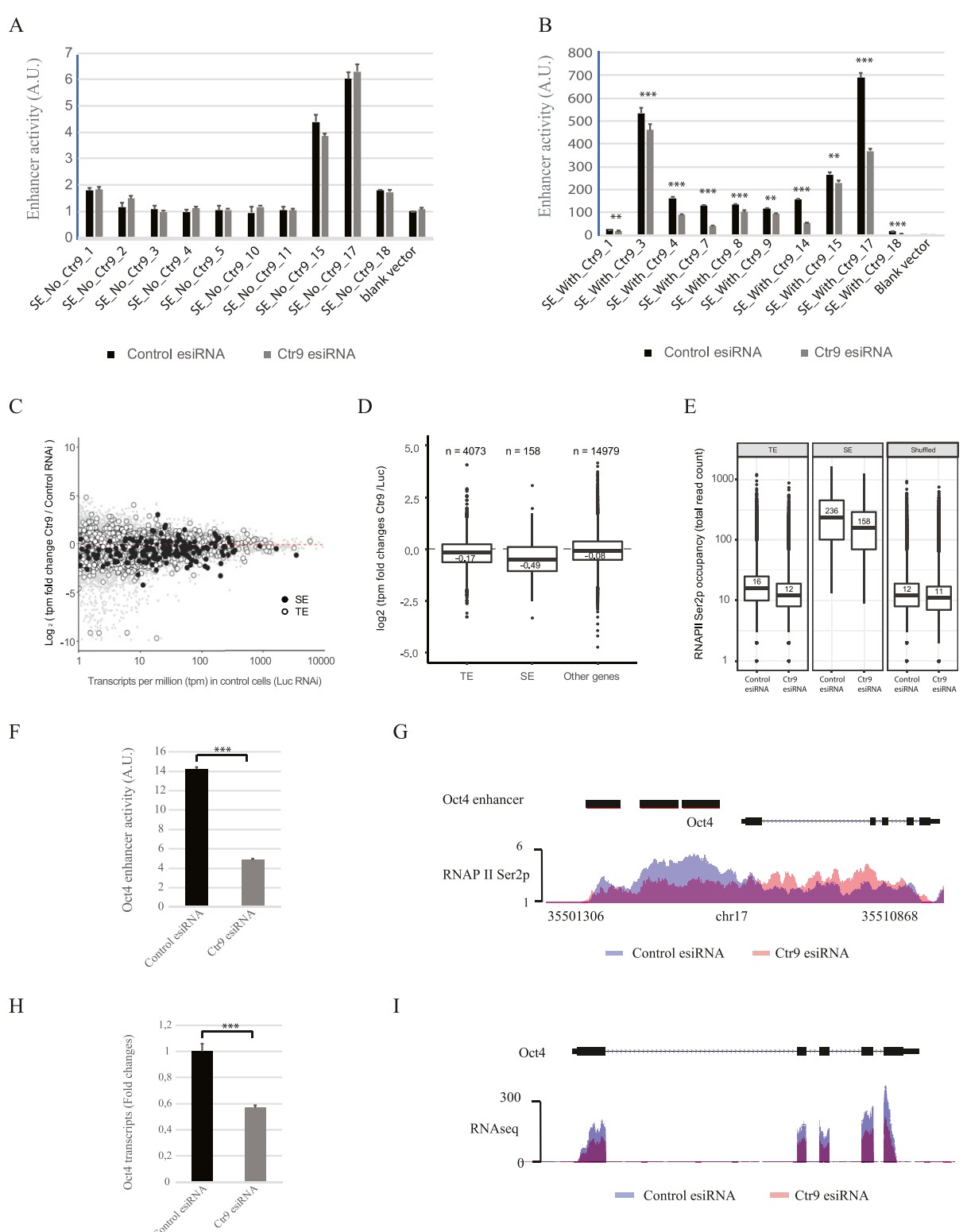

**Figure 5. Paf1C regulates gene expression by modulating enhancer activity.**
**(A)** Enhancer activity of indicated super enhancer (SE) sequences not bound by Ctr9 after transfection with a non-targeting control silencing trigger (black) or Ctr9 knockdown (grey) are shown. **(B)** Enhancer activity of indicated SE sequences that are bound by Ctr9 after transfection with a non-targeting control silencing trigger (black) and Ctr9 knockdown (grey) are shown. The y-axis shows the luciferase measurements in A.U. The values are normalized to samples transfected with the empty reporter vector control. The pRL-SV40 plasmid was used as transfection efficiency control. Data are presented as the mean ± SD from three independent experiments. Error bars depict SD. **(A, B)** Note that depletion of Ctr9 strongly reduced enhancer activity of SEs with Ctr9 binding (panel B), whereas the same treatment had no significant

of active enhancers for future studies. To systematically investigate the role of Paf1C at enhancers in the future, it would be interesting to compare the presented Ctr9 ChIP-seq data with STARR-seq experiments carried out in mESCs.

# Materials and Methods

### Cell culture and esiRNA transfection

mESCs E14TG2a (E14) and NIH3T3 cells were cultured in Glasgow Minimum Essential Medium (GMEM, G5154; Sigma-Aldrich), supplemented with 10% FBS (2602-P290705; Pan-Biotech), 1,000 U/ml LIF (ESG1106; ESGRO), 100 mM nonessential amino acids (11140-050; Invitrogen), 2 mM L-glutamine (25030-081; Invitrogen), and 50 $\mu$M 2-mercaptoethanol (31350-010; Invitrogen).

Endoribonuclease-prepared siRNAs (EsiRNAs) were produced for gene knockdown as described previously (46). EsiRNA transfection was performed using 2 $\mu$l Lipofectamine 2000 (11668-019; Invitrogen), 800 ng esiRNA, and 80,000 cells per well in six-well plates. Primer sequences for esiRNA production are listed in Table S10.

### CRISPR/Cas9-assisted GFP tagging

Short guide RNA and donor repair template construct (GFP with 800 bp homologous arms on both sides to Ctr9 C-terminus cutting site) were cloned as previous reported (47). Oligonucleotides for plasmid constructions are listed in Table S10. CRISPR/Cas9 plasmid pX459 (62988; Addgene), short guide RNA and donor repair template were co-transfected into mESCs, and NIH3T3 cells. 72 h after transfection, GFP-positive cells expressing Ctr9-GFP fusion protein were sorted and collected by FACSAria (BD Bioscience). To avoid clonal variability, we used a pool of GFP-expressing cells instead of single clones to perform ChIP experiments.

### Enhancer reporter assays

Enhancer fragments were cloned into the KpnI site of the pGL4.23 vector. mESCs and NIH3T3 were transfected with pGL4.23 enhancer reporter plasmids containing designated sequences using Lipofectamine 2000

(Invitrogen). The pRL-SV40 plasmid (Promega) was co-transfected as a normalization control. Cells were incubated for 24 h, and luciferase activity was measured using the Dual-Luciferase Reporter Assay System (Promega). For knockdown experiments, esiRNA was transfected into mESCs using Lipofectamine 2000. mESCs were fed with fresh medium and transfected with pGL4.23 enhancer reporter plasmid and pRL-SV40 using Lipofectamine 2000. The genomic coordinates of the cloned fragments and the primers used are listed in Table S10.

### ChIP-seq

ChIP-seq experiments were performed as previously described with minor modification (48). 10 million cells were fixed with 1% (vol/vol) formaldehyde (methanol free) at room temperature for 10 min. Cross-linked chromatin was prepared and sheared using truChIP Chromatin Shearing Kits (Covaris). Antibodies against H3K27ac (8173S; Cell Signaling), RNAPII Ser2p (ab5095; Abcam), RNAPII Ser5p (13523S; Cell Signaling), and GFP (MPI-CBG) were used for ChIP experiments. ChIP-Seq experiments were performed as single runs.

Sequencing was performed on an Illumina HiSeq 2000, aiming for ~30 million sequencing reads per sample. Obtained sequences were aligned to the mouse genome (ver. mm10) (49) using bowtie2 aligner (ver. 2.3.5.1) (50), using default parameters. Successfully aligned reads were filtered to remove sequences mapped to problematic regions using the ENCODE (51) blacklist downloaded from https://github.com/ Boyle-Lab/Blacklist. The number of sequenced fragments aligning to each base was normalized to the total number of reads in a sample. This normalization step was performed automatically by the peak calling software. Peak calling was performed using macs2 (ver. 2.2.6) (52), using a q-value cutoff 0.01. Annotation of peaks, as well as integration with RNA-Seq data, was performed with HOMER software package (ver. 4.10.1) (53) and BEDTools (ver. 2.29.2) (54). Heat maps, signal density curves, and box plots were generated in R (https:// www.R-project.org, version 3.5), using standard methods from tidyverse packages (55). Enhancer occupancy by Ctr9, H3K4me1, H3K27ac, Ser5P, and Ser2P was determined by detection of an at least 1 bp intersection between genomic coordinates of an enhancer region and a respective ChIP-Seq peak. Distances between peaks were measured as distances between predicted precise binding sites, which are defined as locations with the highest fragment pileup within a peak location (peak summits), as reported by MACS2.

---

effects on SEs without Ctr9 binding (panel A). **(C)** MA plot showing global gene expression changes upon Ctr9 knockdown, with each point representing one transcript. White-filled and black-filled circles indicate transcripts associated with typical enhancers (TEs) or SEs, respectively. Grey dots indicate all other transcripts that are not associated with SEs or TEs. A dashed red line shows a boundary at which gene expression is not altered. The y-axis shows a $\log_2$ fold change of transcript per million (tpm) values measured in cells treated by Ctr9 RNAi versus control. Only genes considered as expressed (tpm ≥ 1) are shown. **(D)** Box plots showing distributions of gene expression changes upon RNAi-based Ctr9 knockdown. Genes associated with different types of enhancers are split into separate groups (labelled as TE and SE) and the third group, labelled "Other genes," contains genes not associated with any annotated enhancer. The y-axis shows a $\log_2$ fold change of tpm values measured in cells treated by Ctr9 RNAi versus control. $P$-value < $2.58 \times 10^{-05}$ (SEs versus TEs) and $P$-value < $1.7 \times 10^{-09}$ (SEs versus "Other genes") were calculated according to Mann–Whitney U test. **(E)** Box plots showing general reductions of Pol II Ser2p occupancy at enhancers upon Ctr9 knockdown. The first two panels show total read counts across 8,563 typical enhancers (TEs) and 231 super enhancers (SEs). The third panel, labelled "Shuffled," serves as a control showing data from 8,794 randomized genomic intervals of the same length as in both sets of enhancers. Numbers above horizontal bars are sample medians. **(F)** Enhancer activity of *Oct4* SE after transfection with a non-targeting silencing trigger control (black) or Ctr9 knockdown (grey) is shown. The y-axis shows the luciferase measurements in A.U. **(G)** RNAPII Ser2p occupancy on *Oct4* SEs after transfection with a non-targeting silencing trigger control (blue) or Ctr9 knockdown (red) is shown. The x-axis indicates the chromosome position, and the y-axis represents normalized read density in reads per million. Black boxes indicate the annotated SEs. Each ChIP-Seq experiment was performed with a single sample. **(H)** qRT-PCR quantification of *Oct4* expression after transfection with a non-targeting silencing trigger control (black) or Ctr9 knockdown (grey). *Oct4* expression was normalized to the expression of the housekeeping gene GAPDH. Values and shown as fold changes to the sample transfected with non-targeting silencing trigger control. Data are presented as the mean ± SD from three independent experiments. Error bars depict SD. **(I)** *Oct4* expression determined by RNA-seq after transfection with a non-targeting silencing trigger control (blue) or Ctr9 knockdown (red). The y-axis represents normalized read density in reads per 10 million. Statistically significant differences were determined by a two-tailed $t$ test (** indicates $P < 0.01$ and *** indicates $P < 0.001$).

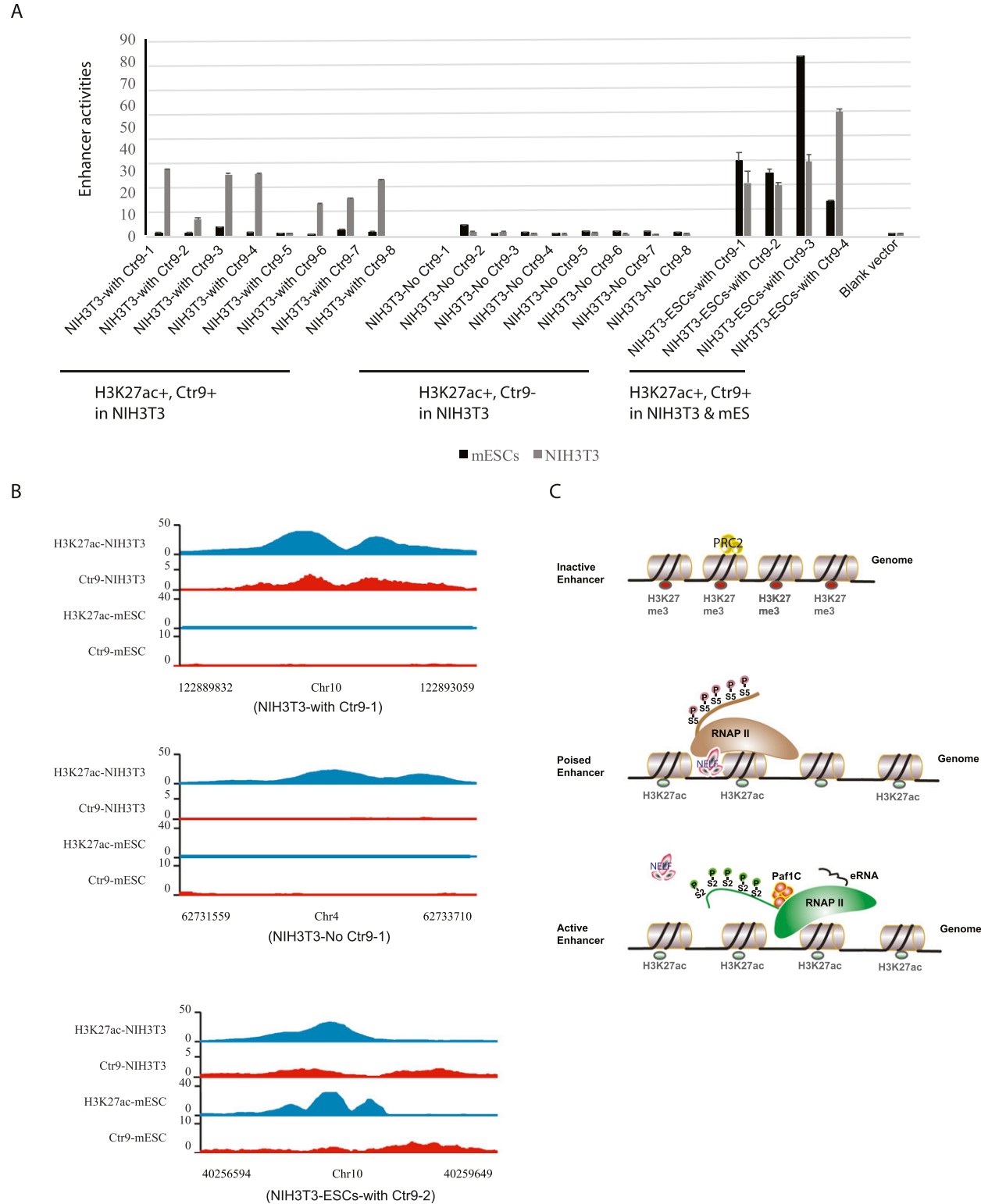

**Figure 6.  The combination of H3K27ac and Ctr9 DNA-occupancy improves the prediction of active enhancers.**
**(A)** Enhancer activities of indicated regions with H3K27ac+/Ctr9+ or H3K27ac+/Ctr9− marks in NIH3T3 cells (grey) and mouse embryonic stem cells (black). The y-axis shows the luciferase measurements in A.U. The values are normalized to samples transfected with the empty reporter vector. The pRL-SV40 plasmid was used as transfection efficiency control. Data are presented as the mean ± SD from three independent experiments. Error bars depict SD. Chromosome positions of each indicated region are listed in Table S9. **(A, B)** Representative genome browser views of regions that were analyzed for enhancer activity in panel (A). Genome sites with H3K27ac+/Ctr9+ mark in NIH3T3 cells (upper panel), H3K27ac+/Ctr9− marks in NIH3T3 cells (middle panel), and H3K27ac+/Ctr9+ marks in both NIH3T3 and mES cells (lower panel) are shown. **(A)** The x-axis indicates the chromosome position of the corresponding fragments analyzed in panel (A). The y-axis represents normalized read density in reads per million. Each ChIP-Seq experiment was performed with a single sample. **(C)** Model of indicated states of enhancer activity. Proposed roles of indicated protein complexes and chromatin modifications are depicted. Polycomb repressive complex 2, PRC2.

### qRT-PCR and RNA-seq

Total RNA was extracted from cells using RNeasy Mini Kit (QIAGEN). 500 ng total RNA was reversed transcribed with Superscript III Reverse Transcriptase (Invitrogen) using an oligo d(T)$_{18}$ primer. qPCR quantification of gene knockdown was performed with the SYBR Green qPCR kit (Abgene) with primers as shown in Table S10.

RNA-Seq was performed on an Illumina HiSeq 2000, aiming for ~30 million sequencing reads per sample. Expression levels of individual transcripts were estimated by kallisto (ver. 0.46.1) (56) using GENCODE transcriptome (release M19) (57) as a reference. For all gene-level comparisons, only representative transcripts with the highest expression values per gene were used.

### Immunostaining

Immunostaining was performed as previous reported (58). Briefly mESCs (Ctr9-GFP, NELFA-GFP) were cultured on cover slips. 4% PFA was used to fix the cells for 10 min at room temperature. After washed with PBS for three times, blocking and permeabilization buffer (1% BSA and 0.5% Triton X-100 in PBS) was used to permeabilize the cells for 15 min at room temperature. Cells were then incubated with primary antibodies at 4°C overnight and then with fluorescent probe-conjugated secondary antibodies for 1 h at room temperature. DAPI (62248; Thermo Fisher Scientific) was used to stain Nuclei at room temperature for 5 min.

### Western blot hybridization

$1.5 \times 10^5$ mES cells (Ctr9-GFP, NELFA-GFP) were reverse transfected with 1,000 ng esiRNAs and 2 $\mu$l lipofectamine 2000 in fibronectin-coated six-well plates. 72 h post transfection, ESCs cells were harvested and lysed in Laemmli sample buffer. 10 $\mu$g of protein extracts were separated on NuPAGE 4–12% Bis-tris protein gels (Invitrogen) and blotted to nitrocellulose membrane (Millipore). The membranes were probed with the primary antibodies against anti-GFP (MPI-CBG), and Tubulin (sc-58880; Santa Cruz), and corresponding secondary antibodies (RDye 680RD/800CW anti-mouse IgG, anti-rabbit IgG, and anti-goat IgG). The membranes were scanned by an Odyssey Infrared Imager, and the proteins were quantified by the software Image Studio.

## Data Availability

The supporting data sets including ChIP-seqs and RNAseqs have been deposited in the Gene Expression Omnibus database (http://ncbi.nlm.nih.gov/geo) with accession number GSE149999.

## Supplementary Information

## Acknowledgements

We thank Andreas Dahl (Dresden-concept Genome Center) for assistance with ChIP-seq and RNA-seq experiments. We are grateful to Katja Bernhardt, FACS facility of the Center for Regenerative Therapies Dresden (CRTD), Technische Universität Dresden for excellent technical assistance in cell sorting. This work was supported by the European Union FP7 grant SyBoSS (242129) and a grant by the Deutsche Forschungsgemeinschaft (BU 1400/5-2).

### Author Contributions

L Ding: investigation and writing—original draft.
M Paszkowski-Rogacz: data curation, software, and methodology.
J Mircetic: conceptualization, data curation, and methodology.
D Chakraborty: investigation and methodology.
F Buchholz: conceptualization, data curation, supervision, funding acquisition, project administration, and writing—review and editing.

### Conflict of Interest Statement

The authors declare that they have no conflict of interest.

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
