## [Reviewer comments · Life Science Alliance]

Life Science Alliance

The Paf1 complex positively regulates enhancer activity in mouse embryonic stem cells

Li Ding, Maciej Paszkowski-Rogacz, Jovan Mircetic, Debojyoti Chakraborty, and Frank Buchholz
DOI: <https://doi.org/10.26508/lsa.202000792>

Corresponding author(s): Frank Buchholz, Medical Systems Biology

Review Timeline:	Submission Date:	2020-05-25
	Editorial Decision:	2020-06-17
	Revision Received:	2020-11-11
	Editorial Decision:	2020-12-07
	Revision Received:	2020-12-17
	Accepted:	2020-12-18

Scientific Editor: Shachi Bhatt

Transaction Report:

June 17, 2020

Re: Life Science Alliance manuscript #LSA-2020-00792-T

Dr. Frank Buchholz
UCC - University Cancer Center Dresden
Medical Systems Biology, UCC, Medical Faculty Carl Gustav Carus
Dresden
Germany

Dear Dr. Buchholz,

Thank you for submitting your manuscript entitled "The Paf1 complex positively regulates enhancer activity in mouse embryonic stem cells" to Life Science Alliance.

Please excuse the delay in communicating this decision to you which was also due to delayed referee reports on account of the current pandemic situation. We have now however received three reports on your study which are included below for your information.

As you will see, the referees appreciate an overall interest in further characterizing the link between the PAF complex and enhancers, but also raise several concerns that would need to be addressed in a revised manuscript. In particular, referee #1's point 5 should be addressed by experimentally demonstrating that the effect is specific and gene expression not globally downregulated. In addition, please provide the experimental controls referee #2 indicates in point 1, 3, 8, and importantly, clarifying the experimental approach for Figure 6 (ref#2- point 11).

Furthermore, both referee #1 and #2 find that the conclusion that NELF1 is displaced by PAF1C is not sufficiently supported by the data provided and this statement should be revised to accurately reflect the experimental evidence (ref#1- point 1, ref#2- point 5). Please also revise the manuscript text to reflect the concerns, as well as adding the requested additional information on experimental procedures, data analysis and statistics for the following points: ref#1 - points 2, 3, 6, 7, 8, ref#2- points 2, 4, 5, 10 and ref#3 points 2, 3, 4. Moreover, the data presentation should be further expanded and a metagene plot (ref#1- point 1) and heatmaps (ref#2- point 6, 7, 9) provided as indicated. In addition, please also ensure that the current study is carefully discussed in the context of previous work (ref#2 - point 4, ref#- point 1).

If you find that you will be able to adequately address these issues, we would be happy to consider the study further for publication in Life Science Alliance. Therefore, we would now like to invite you to prepare and submit a revised version.

We realize that lab work worldwide is currently affected by the COVID-19/SARS-CoV-2 pandemic and that an experimental revision may be delayed or temporarily impossible. The typical timeframe for revisions is three months, but we can extend the revision time when needed, and we have extended our 'scooping protection policy' to cover the period required for a full revision. However, it is nonetheless important to clarify any questions and concerns at this stage and we encourage you to discuss a revision plan and any potential issues you may foresee as soon as possible.

Please note that papers are generally considered through only one revision cycle, so strong support from the referees on the revised version is needed for acceptance.

Thank you for this interesting contribution to Life Science Alliance. We are looking forward to receiving your revised manuscript.

Sincerely,

Reilly Lorenz
Editorial Office Life Science Alliance
Meyerhofstr. 1
69117 Heidelberg, Germany
t +49 6221 8891 414
e contact@life-science-alliance.org
www.life-science-alliance.org

B. MANUSCRIPT ORGANIZATION AND FORMATTING:

Reviewer #1 (Comments to the Authors (Required)):

Ding et al. examined the genome-wide occupancy of Ctr9, a subunit of the elongation factor Paf1C, and NELFA, a subunit of the negative elongation factor NELF, in mouse embryonic stem cells. To this end they tagged endogenous Ctr9 and NELFA proteins with GFP utilizing genome editing with CRISPR/Cas9. The authors show that Ctr9 is associated with protein coding genes and the occupancy correlates with expression levels. They next examined Ctr9 occupancy at enhancers and super enhancers, and show that Ctr9-occupancy is linked to enhancer activity in a plasmid reporter assay. The authors knock down Ctr9 with siRNA to show that it is needed for enhancer activity. The knockdown reduced the activity of Ctr9-bound enhancers in the reporter assay and was further supported by two examined endogenous enhancers, where Ctr9 knockdown reduced RNAPII-Ser2P occupancy at Oct4- and Tbx3-enhancers and transcription at the corresponding genes. The authors indicate that Ctr9-occupancy could be a marker of active enhancers and substantiate their claim, by performing Ctr9 ChIP-seq for NIH3T3 cell line and the enhancer reporter assay with a subset of enhancers in the same cell line. Based on these results the authors propose that Paf1C occupancy could be utilized to classify active enhancers and that Paf1C is required for enhancer function. More specifically, that Paf1C activity at pluripotency enhancers is required for maintenance of stem cell self-renewal.

The authors make several statements based on weak or lacking evidence. In particular, in the abstract the authors claim that Ctr9 regulates gene expression by regulating the activity of enhancers, specifically pluripotency enhancers crucial for maintenance of mESC self-renewal. This is unsubstantiated and no evidence for specificity is shown (for example evidence that other genes are not affected). Also, the authors claim that they have shown "that the Paf1C positively regulates RNAPII pause-release at both protein-coding genes and at enhancers in mouse embryonic stem cells (mESC)", but provide no experimental quantification or analyses of pause-release. The Ctr9 ChIP-seq data analyses appear to be oversimplified and overinterpreted. Furthermore, the authors provide insufficient and questionable evidence to support their claim that Paf1C displaces NELF for active transcription of protein coding genes. Additionally, the authors tend to show one example and make global or wide-sweeping statements. Lastly, the methods section is lacking critical information and there is no indication that the ChIP-seq experiments were performed with replicates. Therefore I cannot recommend this study for publication. I hope the comments below help the authors to obtain an improved manuscript.

1. "Paf1C displaces NELF for active transcription of protein coding genes" - the statement is based

on overinterpretation of ChIP-seq beyond the resolution of the method. 1E is a Gaussian kernel density distribution of ChIP-seq peaks. This is misleading, firstly, the resolution of ChIP-seq is approximately the size of the DNA fragments (200-500 bp), second, the peaks can be broad or narrow peaks - the midpoints do not represent the actual occupancy. A metagene profile for the actual coverage should be shown instead.

2. Figure 3B. It is unclear how was this distance measured - midpoint to midpoint or end to start of the peak range? In the heatmap there appears to be more Ctr9 bound in the window shown in the heatmaps in 3A for SE than for TE and also more overlapping intensities, which contradicts the information shown in 3B.

3. "In addition to protein coding genes, we found that RNAPII Ser5p and Ser2p were also significantly enriched on SEs, with 83% and 63% occupancy, respectively, suggesting that a majority of SEs are transcriptional enhancers (Dataset EV4, Fig 4A, 4B)." How did the authors get these numbers, also no data to support this in the referred figure panels. SEs are generally considered to be transcriptional enhancers.

4. "Upon Ctr9 depletion, 53% of the SEs exhibited decreased RNAPII Ser2p occupancy,..." data supporting the authors claim is not shown.

5. "Strikingly, SEs with a strong decrease of RNAPII Ser2p after Ctr9 knockdown lost their enhancer activity dramatically, and this effect correlated with a strong decrease in transcription of their associated target genes, such as Oct4 and Tbx3 (Fig 5C-5F; Fig EV5A-5D), suggesting that Paf1C regulates gene expression by modulating enhancer activity." For the authors to make this claim, they would need to show that this is specific and not a global gene expression down-regulation upon Ctr9 knockdown and that this is not only valid for Oct4 and Tbx3 SE.

6. "... we have performed a panel of experiments to determine DNA-binding of relevant factors, including NELFA, RNAPII Ser5p, RNAPII Ser2P and Ctr9 in mESCs." The authors performed ChIP-seq, this experiment does not prove direct binding to DNA.

7. "Our analyses are consistent with a dynamic transition of NELF, RNAPII Ser5p and Paf1C within a 150-bp region downstream of the TSS, indicating a coordinated action of events of these factors for active transcription in mESCs, where NELF is substituted by Paf1C to release RNAPII from promoter proximal pausing." The supposed distances between NELFA and Ctr9 peaks is 354 and 465 for TEs and SEs, respectively. This contradicts the concept that this transition would occur within a 150-bp window.

8. Incorrect statements: "...exchange of RNAPII-Ser5p for Paf1C...", "NELFA is an initiation factor"; "... Paf1C binding can be used in combination with histone marks to predict active enhancers in primary cells." - NIH3T3 are not primary cells.

Reviewer #2 (Comments to the Authors (Required)):

In this manuscript, the authors investigate the connection between the PAF complex and enhancer function in mouse ES cells. Using ChIP-seq experiments on PAF subunit Ctr9 and reporter assays, the authors make several important discoveries. First, they show that Ctr9 occupies many but not all enhancers in a cell-type specific manner. Second, they show that Ctr9 occupancy correlates with

transcription at the enhancer and the presence of eRNAs. Third, they show that Ctr9-occupied enhancers, but not those lacking Ctr9, activate transcription in their reporter assay. Further, depletion of Ctr9 reduces transactivation by these enhancers and Pol II Ser2p levels at the enhancer, suggesting PAF controls transcription at the enhancer. Finally, they provide evidence that Ctr9 occupancy, together with H3K27ac, is a predictor of enhancer activity. What is left unanswered by this work is what determines whether an enhancer is occupied by Ctr9 (and hence active) versus unoccupied (and hence inactive). Nonetheless, the contributions will be of interest to many in the enhancer field, as there is considerable debate about the role of PAF in enhancer function and Pol II pausing. However, in its current form, there are weaknesses in experimental design, data analysis and presentation that need to be addressed.

Major comments:

1. The paper requires a more rigorous description of reproducibility and controls. For the GFP-tagging of Ctr9 and NELFA, how many independently tagged mESC and NIH3T3 cell lines were generated and tested? Given the possibility of CRISPR-generated indirect and off-target effects, a test of more than one independently derived line would substantiate the claims made in the paper regarding Ctr9 and NELF occupancy. Does the GFP tag on either protein affect occupancy? The authors should consider confirming their results with unaltered mESCs and antibodies against endogenous Ctr9 and NELF to show that the GFP tags are not having an influence and that CRISPR off-target effects are not impacting the data.
2. Regarding the ChIP-seq experiments, the methods do not describe a spike-in control or replicate numbers. The methods for normalizing ChIP-seq signals and data reproducibility (number of replicates, correlations between replicates) need to be stated.
3. The methods do not adequately describe how the GFP-tagged cell lines were confirmed or characterized, other than showing that the tagged proteins were expressed and nuclear.
4. Figure 1A,B. These data are consistent with substantial existing literature showing a strong correlation between gene expression (and Pol II occupancy) and PAF enrichment. Although the data were presented, the authors did not address the strong Ctr9 ChIP-seq signal downstream of the TTS. This should be discussed in the context of work by Yang et al. (PLOS Genetics 2016) who noted this downstream enrichment in mouse myoblasts and showed an effect of PAF on polyA site usage.
5. Figure 1D,E,F. The authors report the separated ChIP peaks for NELFA, Pol II Ser5p, and Ctr9. The authors arrive at a conclusion of factor exchange that is well supported by earlier studies. The data are consistent with recent structures from the Cramer lab, which showed that binding of NELF and PAF to Pol II is mutually exclusive. The Cramer lab also reported in 2010 that PAF is enriched downstream of Ser5p. While the authors' ChIP-seq results provide nice support for these results in mESCs, they do not address mechanism and are largely confirmatory. The title of this figure, "Paf1C displaces the NELF complex during promoter proximal pause release of target genes", is overstated. In addition, from the box plot in Figure 1F, it seems a significant number of genes show an overlap between the position of Ctr9 occupancy and the position of NELFA. An explanation for this observation is needed.
6. Figure 2. The occupancy data support previous work from the Shilatifard group, using HCT116 cells, showing PAF localization at enhancers. However, the advance here is the focus on ESCs, pluripotency genes and super enhancers. The demonstration that some super enhancers are unoccupied by Ctr9 and these are deficient in enhancer activity in a luciferase assay is an important result, as it counters current literature from the Shilatifard lab arguing that PAF negatively regulates enhancer function. My main suggestion to improve this figure and the paper in general is to provide genome-wide heatmaps. The paper is heavily reliant on metaplots and browser tracks, either of which can provide a skewed view of the data. What is the occupancy pattern of Ctr9, H3K27ac, and H3K4me1 at all proposed enhancers?

7. Figure 3. The data show distinct occupancy patterns for NELFA and Ctr9 at enhancers (both super and typical). Why are so few examples shown? The authors mention looking at 166 super enhancers but fewer appear to be shown in the heatmap. Based on the heatmap, the super enhancer in Panel D seems like an exception with respect to the patterns of NELFA and Ctr9 occupancy, making it unclear why it is emphasized or how frequent this pattern was observed.
8. Figure EV3. A loading control needs to be added to the western blot.
9. Figure 4 makes the point that Ctr9 occupancy and transcription of enhancers, as measured by Pol II phosphorylation or eRNA accumulation (from published GRO-seq data), are positively correlated. This is another example of where heatmaps would substantiate the more limited information provided by browser tracks and Venn diagrams. Also, given the binding of PAF to Pol II, the result of correlated PAF occupancy and active transcription is not especially surprising, though showing this result at enhancers is a notable advance.
10. Figure 5. The data show that Ctr9-occupied enhancers are more active than non-occupied enhancers in a reporter assay and enhancer activity is partially Ctr9-dependent. The data also show that depletion of Ctr9 reduces Pol II Ser2p, a measure of transcription at the enhancer. These are among the most significant advances in the paper. However, the authors need to be more accurate in the description of the data. Transactivation by the Ctr9-occupied enhancers is reduced 2-fold or more for only 4/10 of the tested enhancers upon Ctr9 depletion. The authors should soften their conclusions concerning the strength of the effects. In addition, for the reduction in Ser2p levels, the authors should show the generality of the data and not rely solely on browser tracks. Finally, while the reduction in genic transcription upon Ctr9 depletion correlates with enhancer inactivation, it is also possible that loss of PAF is affecting transcription through the gene more directly. This should be considered.
11. Figure 6 tests the hypothesis that H3K27ac and Ctr9 occupancy, together, predict enhancer activity better than H3K27ac. This is potentially an important discovery; however, the experiment is not acceptable as shown. There is no evidence of repetition or statistical significance.

Minor comments:

1. Last sentence on p. 6: "Consistent with its role as an elongation factor, Ctr9 binding exhibited cell-type specificity". The two thoughts in this sentence are not logically connected. Consider modifying to state: "Consistent with its role as an elongation factor associated with active gene transcription,".
2. Figure EV3. The term "LAP-tagging" is confusing.
3. The Chen et al reference is incomplete.
4. Sentence on p. 10 requires clarification. "Interestingly, the overlap of Ctr9 occupancy with TEs carrying H3K27ac modification (8%) was significantly lower than that in SEs (68%), suggesting that Ctr9 predominantly associates with active enhancers (Fig 2C, 2D, Dataset EV3)." How do the authors arrive at the main conclusion (ie. Ctr9 predominantly associates with active enhancers) from the lower occupancy overlap between Ctr9 and H3K27ac at TEs compared to SEs?
5. P. 12. The phrase "suggesting that a majority of SEs are transcriptional enhancers" is confusing. Why are they called super enhancers if they are not transcriptional enhancers? The sentence needs to be written more clearly.
6. Figure 6 title. Rephrase to avoid stating Ctr9 DNA-binding. The factor precipitated need not bind DNA directly.

Reviewer #3 (Comments to the Authors (Required)):

In the study "The Paf1 complex positively regulates enhancer activity in mouse embryonic stem

cells", Ding and colleagues reported functions of Paf1C in regulating enhancer activity, in particular in super enhancers. These results suggest the potential power of Paf1c in mECS self-renewal. This study is important to enlarge our knowledge on enhancer activity. This reviewer has several comments which may be addressed.

1. Previous studies (e.g., Hou et, al., PNAS 2019, Chen et.al., Genes & Development 2009) reported the importance of Paf1C in modulating Pol II elongation. This review suggests that comparison and discussion about previous studies is needed.
2. This review didn't observe characterization of super enhancer in each cell line. Could author describe the method they used in super enhancer definition? Some cases, e.g., figure 2e, figure 3c-d, are unique enhancers in the region, which look like typical enhancer.
3. Author claimed that H3K27ac plus Ctr9 binding could improve prediction of activated enhancer. This reviewer agrees with this point by some way. While as author introduce in the manuscript, transcript enhancer (also known as enhancer RNA) could be a golden standard for active enhancer. Could author consider this important signal in definition of active enhancer?
4. H3K27ac is the marker for both active enhancer and promoter. If the author focus on enhancer only, other histone markers, e.g., H3K4me1, p300, are needed.

Reviewer #1 (Comments to the Authors (Required)):

Ding et al. examined the genome-wide occupancy of Ctr9, a subunit of the elongation factor Paf1C, and NELFA, a subunit of the negative elongation factor NELF, in mouse embryonic stem cells. To this end they tagged endogenous Ctr9 and NELFA proteins with GFP utilizing genome editing with CRISPR/Cas9. The authors show that Ctr9 is associated with protein coding genes and the occupancy correlates with expression levels. They next examined Ctr9 occupancy at enhancers and super enhancers, and show that Ctr9-occupancy is linked to enhancer activity in a plasmid reporter assay. The authors knock down Ctr9 with siRNA to show that it is needed for enhancer activity. The knockdown reduced the activity of Ctr9-bound enhancers in the reporter assay and was further supported by two examined endogenous enhancers, where Ctr9 knockdown reduced RNAPII-Ser2P occupancy at Oct4- and Tbx3-enhancers and transcription at the corresponding genes. The authors indicate that Ctr9-occupancy could be a marker of active enhancers and substantiate their claim, by performing Ctr9 ChIP-seq for NIH3T3 cell line and the enhancer reporter assay with a subset of enhancers in the same cell line. Based on these results the authors propose that Paf1C occupancy could be utilized to classify active enhancers and that Paf1C is required for enhancer function. More specifically, that Paf1C activity at pluripotency enhancers is required for maintenance of stem cell self-renewal.

Response: We thank reviewer#1 for the constructive criticism and suggestions that we have addressed as indicated for each point.

The authors make several statements based on weak or lacking evidence. In particular, in the abstract the authors claim that Ctr9 regulates gene expression by regulating the activity of enhancers, specifically pluripotency enhancers crucial for maintenance of mESC self-renewal. This is unsubstantiated and no evidence for specificity is shown (for example evidence that other genes are not affected). Also, the authors claim that they have shown "that the Paf1C positively regulates RNAPII pause-release at both protein-coding genes and at enhancers in mouse embryonic stem cells (mESC)", but provide no experimental quantification or analyses of pause-release

Response: We have further strengthened the claim that Ctr9 regulates gene expression by regulating the activity of enhancers. We had already shown specificity for Ctr9 bound and non-bound enhancers in Fig. 5A and B. We now add an example of a gene not affected (e.g. Zfp638, Fig. EV5E-EV5H). Furthermore, we have added a global analysis of transcriptional changes of genes regulated by ESC SEs, TEs and other genes (Fig. 5C, 5D). These data substantiate the notion that Ctr9 regulates gene expression by regulating the activity of enhancers.

We agree with the reviewer that we do not show direct evidence for Paf1C-mediated RNAPII pause-release. We have therefore re-phrased the statement in the revised manuscript (Page 8).

The Ctr9 ChIP-seq data analyses appear to be oversimplified and overinterpreted. Furthermore, the authors provide insufficient and questionable evidence to support their claim that Paf1C displaces NELF for active transcription of protein coding genes

Response: We have performed additional analyses of the data, and now provide metagene plots and heatmap analyses to better interpret the ChIPseq data in the revised manuscript (Figure 1E, 3A, 5E). We agree with the reviewer that the displacement of NELF by Paf1C is not directly shown by our data. We have therefore revised our interpretation of NELF1-Paf1C displacement in the revised manuscript (Page 8).

Additionally, the authors tend to show one example and make global or wide-sweeping statements

Response: We apologize if we did not emphasize the large-scale analyses of our data-sets sufficiently. To increase global statements, we have now performed additional experiments and carried out in-depth heatmap and metagene plot analyses. New result and global data analyses are now shown in the revised manuscript (Figures 3A, 5C, 5D).

Lastly, the methods section is lacking critical information and there is no indication that the ChIP-seq experiments were performed with replicates

Response: Missing information has been added to the methods section in the revised manuscript. ChIP-seq experiments were not performed in replicates. However, we would like to point out that the GFP antibody we used is of very high quality and that we have confirmed the GFP-tagging approach in numerous publications, many of which at very high throughput (e.g (Ding et al., 2009; 2015; Hein et al., 2015; Jahn et al., 2017; Kappei et al., 2013; Nitzsche et al., 2011; Poser et al., 2008)). We are therefore confident that the presented data is of high quality.

Therefore I cannot recommend this study for publication. I hope the comments below help the authors to obtain an improved manuscript.

1. "Paf1C displaces NELF for active transcription of protein coding genes" - the statement is based on overinterpretation of ChIP-seq beyond the resolution of the method. **1E** is a Gaussian kernel density distribution of ChIP-seq peaks. This is misleading, firstly, the resolution of ChIP-seq is approximately the size of the DNA fragments (200-500 bp), second, the peaks can be broad or narrow peaks - the midpoints do not represent the actual occupancy. A metagene profile for the actual coverage should be shown instead.

Response: We appreciate this comment by the reviewer. Based on his/her suggestion we have replaced the plot by a metagene profile of Ctr9, RNAPII Ser5p and Nelfa binding in the revised version of our manuscript (Figure 1E). The conclusion based on this analysis did not change.

2. **Figure 3B.** It is unclear how was this distance measured - midpoint to midpoint or end to start of the peak range? In the heatmap there appears to be more Ctr9 bound in the window shown in the heatmaps in 3A for SE that for TE and also more overlapping intensities, which contradicts the information shown in 3B.

Response: The distances were measured between predicted peak summits, as reported by the peak calling software. The heatmaps show overall distributions and intensity of signals, whereas boxplots show distributions of distances between peak summits, which are approximately at positions of the highest signal intensity. We have modified the heatmaps to indicate positions of peak summits and provide the distance measured details in the methods section (page 20).

3. "In addition to protein coding genes, we found that RNAPII Ser5p and Ser2p were also significantly enriched on SEs, with 83% and 63% occupancy, respectively, suggesting that a majority of SEs are transcriptional enhancers (Dataset EV4, **Fig 4A, 4B**)." How did the authors get these numbers, also no data to support this in the referred figure panels.

Response: We thank reviewer #1 for this comments. We explain how we calculated these numbers in Dataset EV4. In addition, heatmaps of the analyses are now shown in the revised manuscript (Figure 3A).

SEs are generally considered to be transcriptional enhancers.

Response: We have re-phrased this sentence in the revised manuscript; Thank you.

4. "Upon Ctr9 depletion, 53% of the SEs exhibited decreased RNAPII Ser2p occupancy,..." data supporting the authors claim is not shown.

Response: We calculated a total read coverage over super enhancers in the RNAPII Ser2p samples after Ctr9 depletion. The analysis revealed a 1.5-fold depletion of RNAPII Ser2P signal over super-enhancers and a 1.3-fold depletion over typical enhancers. No significant change of occupancy was observed over a shuffled set of genomic intervals. The new analyses are now shown in the revised manuscript (Figure 5D; Dataset EV4).

5. "Strikingly, SEs with a strong decrease of RNAPII Ser2p after Ctr9 knockdown lost their enhancer activity dramatically, and this effect correlated with a strong decrease in transcription of their associated target genes, such as Oct4 and Tbx3 (Fig 5C-5F; Fig EV5A-5D), suggesting that Paf1C regulates gene expression by modulating enhancer activity." For the authors to make this claim, they would need to show that this is specific and not a global gene expression down-regulation upon Ctr9 knockdown and that this is not only valid for Oct4 and Tbx3 SE.

Response: To address this point, we have analyzed the expression of super enhancer and typical enhancer target genes after Ctr9 knockdown. The expression of the majority of SEs target genes decreased after Ctr9 knockdown. In contrast, the expression of genes that are not associated with enhancers did not show this pattern. The new analysis is shown in the revised manuscript (Figure 5C, 5D).

6. "... we have performed a panel of experiments to determine DNA-binding of relevant factors, including NELFA, RNAPII Ser5p, RNAPII Ser2P and Ctr9 in mESCs." The authors performed ChIP-seq, this experiment does not prove direct binding to DNA.

Response: We agree with the reviewer on this point. We have rephrased this sentence in the revised version of the manuscript. The sentence now reads: "... we have performed a panel of experiments to determine DNA occupancies of relevant factors, including NELFA, RNAPII Ser5p, RNAPII Ser2P and Ctr9 in mESCs."

7. "Our analyses are consistent with a dynamic transition of NELF, RNAPII Ser5p and Paf1C within a 150-bp region downstream of the TSS, indicating a coordinated action of events of these factors for active transcription in mESCs, where NELF is substituted by Paf1C to release RNAPII from promoter proximal pausing." The supposed distances between NELFA and Ctr9 peaks is 354 and 465 for TEs and SEs, respectively. This contradicts the concept that this transition would occur within a 150-bp window.

Response: As mentioned in the text already, different mechanisms may account for the different distances between NEFLA and Ctr9 on enhancers and on protein coding genes. The reviewer might have missed this (page 11, bottom of original submission).

1: eRNA transcription is modulated by a panel of specific transcription factors. For instance, MII3 and MII4 are essential for RNAP II accumulation at enhancers to activate transcription. Loss of MII3/4 from enhancers decrease the binding rate of RNAP II and thus eRNA production (Dorighi et al., 2017). Another study has suggested that condensin could modulate the binding of co-repressors or co-activators (like p300 and RIP140) by the recruitment of an E3 ubiquitin ligase to regulate eRNA transcription (Li et al., 2015).

2: Some eRNAs are bidirectionally transcribed, in contrast to the mostly unidirectional transcription of mRNA (Mikhaylichenko et al., 2018).

Therefore, we stand to our statement that the larger distance suggests that enhancer (e)RNA transcription may be orchestrated in a different way than protein coding genes.

8. Incorrect statements: "...exchange of RNAPII-Ser5p for Paf1C...", "NELFA is an initiation factor"; "... Paf1C binding can be used in combination with histone marks to predict active enhancers in primary cells." - NIH3T3 are not primary cells.

Response: We have rephrased these statements in the revised manuscript.

Reviewer #2 (Comments to the Authors (Required)):

In this manuscript, the authors investigate the connection between the PAF complex and enhancer function in mouse ES cells. Using ChIP-seq experiments on PAF subunit Ctr9 and reporter assays, the authors make several important discoveries. First, they show that Ctr9 occupies many but not all enhancers in a cell-type specific manner. Second, they show that Ctr9 occupancy correlates with transcription at the enhancer and the presence of eRNAs. Third, they show that Ctr9-occupied enhancers, but not those lacking Ctr9, activate transcription in their reporter assay. Further, depletion of Ctr9 reduces transactivation by these enhancers and Pol II Ser2p levels at the enhancer, suggesting PAF controls transcription at the enhancer. Finally, they provide evidence that Ctr9 occupancy, together with H3K27ac, is a predictor of enhancer activity. What is left unanswered by this work is what determines whether an enhancer is occupied by Ctr9 (and hence active) versus unoccupied (and hence inactive). Nonetheless, the contributions will be of interest to many in the enhancer field, as there is considerable debate about the role of PAF in enhancer function and Pol II pausing. However, in its current form, there are weaknesses in experimental design, data analysis and presentation that need to be addressed.

Response: We appreciate the constructive evaluation of our work and the thoughtful comments by reviewer #2. We have addressed his/her points as outlined below:

Major comments:

1. The paper requires a more rigorous description of reproducibility and controls. For the GFP-tagging of Ctr9 and NELFA, how many independently tagged mESC and NIH3T3 cell lines were generated and tested? Given the possibility of CRISPR-generated indirect and off-target effects, a test of more than one independently derived line would substantiate the claims made in the paper regarding Ctr9 and NELF occupancy.

Response: To generate GFP-tagged Ctr9 and NELFA cells we used a pool of GFP expressing cells, instead of a single cell line to perform ChIP experiments for Ctr9 and NELF. The pooling approach eliminates clonal variation which is frequently observed for single cell lines. We have added a statement in the methods section to emphasize this point (page 19).

To validate GFP tagging, we sequenced the flanking regions of GFP, and confirmed correct integration of GFP into the mESC genome as designed. GFP tagging was further confirmed on protein level by Western blot hybridization. GFP fusion proteins of expected molecular weight were detected by the anti-GFP antibody. When treating the cells with esiRNAs targeting Ctr9 or NELFA (Figure EV1, EV3), specific reduction of GFP-tagged proteins was revealed, confirming correct and specific GFP tagging.

Does the GFP tag on either protein affect occupancy? The authors should consider confirming their results with unaltered mESCs and antibodies against endogenous Ctr9 and NELF to show that the GFP tags are not having an influence and that CRISPR off-target effects are not impacting the data.

Response: We thank reviewer #2 for this comment. To address this point, we have tried commercial antibodies targeting Ctr9 and NELFA, but the results were unfortunately not satisfactory, probably because the antibodies were not of sufficient quality. We would like to point out that the GFP antibody we used is of very high quality and that we have confirmed the GFP-tagging approach in numerous publications, many of which at very high throughput (e.g. (Ding et al., 2009; 2015; Hein et al., 2015; Jahn et al., 2017; Kappei et al., 2013; Nitzsche et al., 2011; Poser et al., 2008). We are therefore confident that the presented data is of high quality.

2. Regarding the ChIP-seq experiments, the methods do not describe a spike-in control or replicate numbers. The methods for normalizing ChIP-seq signals and data reproducibility (number of replicates, correlations between replicates) need to be stated.

Response: The ChIP-Seq experiments were not done in replicates and we did not use any spike-in controls. However, we again refer to the validated quality of the GFP-tagging approach (see previous point). The number of sequenced fragments aligning to each base was normalized to the total number of reads in a sample. This normalization step was performed automatically by the peak calling software macs2 (ver. 2.2.6). The method for normalizing ChIP-seq signals is now shown in the revised manuscript (page 20).

3. The methods do not adequately describe how the GFP-tagged cell lines were confirmed or characterized, other than showing that the tagged proteins were expressed and nuclear.

Response: Correct GFP-tagging was confirmed by sequencing of the GFP-tagging region from genomic DNA. In addition, the specific depletion of tagged protein is now shown by using specific esiRNAs to deplete the protein (Figure EV1, EV3).

4. **Figure 1A,B.** These data are consistent with substantial existing literature showing a strong correlation between gene expression (and Pol II occupancy) and PAF enrichment. Although the data were presented, the authors did not address the strong Ctr9 ChIP-seq signal downstream of the TTS. This should be discussed in the context of work by Yang et al. (PLOS Genetics 2016) who noted this downstream enrichment in mouse myoblasts and showed an effect of PAF on polyA site usage.

Response: We thank the reviewer for this suggestion. We have extended this aspect in the revised manuscript (page 6) and we reference the suggested manuscript.

5. **Figure 1D,E,F.** The authors report the separated ChIP peaks for NELFA, Pol II Ser5p, and Ctr9. The authors arrive at a conclusion of factor exchange that is well supported by earlier studies. The data are consistent with recent structures from the Cramer lab, which showed that binding of NELF and PAF to Pol II is mutually exclusive. The Cramer lab also reported in 2010 that PAF is enriched downstream of Ser5p. While the authors' ChIP-seq results provide nice support for these results in mESCs, they do not address mechanism and are largely confirmatory. The title of this figure, "Paf1C displaces the NELF complex during promoter proximal pause release of target genes", is overstated. In addition, from the box plot in Figure 1F, it seems a significant number of genes show an overlap between the position of Ctr9 occupancy and the position of NELFA. An explanation for this observation is needed.

Response: We would like to thank the reviewer for the thoughtful comments. We have rephrased the title of this figure.

Due to a broader range of Ctr9 occupancy, a precise estimation of peak summit positions is quite challenging. We would attribute this large spread of distances to a poor accuracy of peak summit positioning.

6. **Figure 2.** The occupancy data support previous work from the Shilatifard group, using HCT116 cells, showing PAF localization at enhancers. However, the advance here is the focus on ESCs, pluripotency genes and super enhancers. The demonstration that some super enhancers are unoccupied by Ctr9 and these are deficient in enhancer activity in a luciferase assay is an important result, as it counters current literature from the Shilatifard lab arguing that PAF negatively regulates enhancer function. My main suggestion to improve this figure and the paper in general is to provide genome-wide heatmaps. The paper is heavily reliant on metaplots and browser tracks, either of which can provide a skewed view of the data. What is the occupancy pattern of Ctr9, H3K27ac, and H3K4me1 at all proposed enhancers?

Response: We have re-analyzed Ctr9, H3K27ac, and H3K4me1 at all studied enhancers, and prepared a heatmap. This analysis is now shown in the revised manuscript (Figure 3A, Dataset EV3).

7. **Figure 3.** The data show distinct occupancy patterns for NELFA and Ctr9 at enhancers (both super and typical). Why are so few examples shown? The authors mention looking at 166 super enhancers but fewer appear to be shown in the heatmap. Based on the heatmap, the super enhancer in Panel D

seems like an exception with respect to the patterns of NELFA and Ctr9 occupancy, making it unclear why it is emphasized or how frequent this pattern was observed.

Response: We thank the reviewer for the suggestion. We have increased the number of enhancers that we show in Figure 3A. To emphasize the relationships between the binding patterns, we decided to show enhancers at which we identified strong binding sites for both NELFA and Ctr9.

8. Figure EV3. A loading control needs to be added to the western blot.

Response: We have repeated the Western blot hybridization in NELFA-GFP cell line. Specific depletion of NELFA-GFP fusion protein was detected after RNAi. Tubulin was used as loading control. The Western blot hybridization is now shown in the modified version of the manuscript (Figure EV3B).

9. **Figure 4** makes the point that Ctr9 occupancy and transcription of enhancers, as measured by Pol II phosphorylation or eRNA accumulation (from published GRO-seq data), are positively correlated. This is another example of where heatmaps would substantiate the more limited information provided by browser tracks and Venn diagrams. Also, given the binding of PAF to Pol II, the result of correlated PAF occupancy and active transcription is not especially surprising, though showing this result at enhancers is a notable advance.

Response: We thanks review #2 for this suggestion. We have re-analyzed RNAPII Ser2P, Ser5P, NELFA, Ctr9 H3K27ac and H3K4me1 occupancy at super enhancers and typical enhancers. The heatmap and analysis are now shown in the revised manuscript (Figure 3A and Dataset EV4).

10. **Figure 5.** The data show that Ctr9-occupied enhancers are more active than non-occupied enhancers in a reporter assay and enhancer activity is partially Ctr9-dependent. The data also show that depletion of Ctr9 reduces Pol II Ser2p, a measure of transcription at the enhancer. These are among the most significant advances in the paper. However, the authors need to be more accurate in the description of the data. Transactivation by the Ctr9-occupied enhancers is reduced 2-fold or more for only 4/10 of the tested enhancers upon Ctr9 depletion. The authors should soften their conclusions concerning the strength of the effects. In addition, for the reduction in Ser2p levels, the authors should show the generality of the data and not rely solely on browser tracks. Finally, while the reduction in genic transcription upon Ctr9 depletion correlates with enhancer inactivation, it is also possible that loss of PAF is affecting transcription through the gene more directly. This should be considered.

Response: We have rephrased and soften our conclusion in the revised manuscript (page 13). Furthermore, we checked Ser2p levels on SEs and gene body after Ctr9 knockdown, and observed a predominant decrease of Ser2p on enhancers, which suggests Paf1C regulate transcription through modulation of enhancer activity (Figure 5C, 5D, 5E). Genome browser tracks of Ser2p over SEs and gene body are shown in the modified manuscript (Figure 5G, Figure EV5B).

11. **Figure 6** tests the hypothesis that H3K27ac and Ctr9 occupancy, together, predict enhancer activity better than H3K27ac. This is potentially an important discovery; however, the experiment is not acceptable as shown. There is no evidence of repetition or statistical significance.

Response: We repeated the experiments with replicates. New data with standard deviation are now shown in the revised manuscript (Figure 6A).

Minor comments:

1. Last sentence on p. 6: "Consistent with its role as an elongation factor, Ctr9 binding exhibited cell-type specificity". The two thoughts in this sentence are not logically connected. Consider modifying to state: "Consistent with its role as an elongation factor associated with active gene transcription,".

Response: We have rephrased this sentence accordingly, thank you.

2. Figure EV3. The term "LAP-tagging" is confusing.

Response: We have changed the term LAP-tagging to "Localization and Affinity Purification tag" in the revised manuscript.

3. The Chen et al reference is incomplete.

Response: We are now providing the complete reference for Chen et al. in the revised manuscript.

4. Sentence on p. 10 requires clarification. "Interestingly, the overlap of Ctr9 occupancy with TEs carrying H3K27ac modification (8%) was significantly lower than that in SEs (68%), suggesting that Ctr9 predominantly associates with active enhancers (Fig 2C, 2D, Dataset EV3)." How do the authors arrive at the main conclusion (ie. Ctr9 predominantly associates with active enhancers) from the lower occupancy overlap between Ctr9 and H3K27ac at TEs compared to SEs?

Response: Interestingly almost all Ctr9 binding SEs and TEs carry H3K27ac modification, suggesting that Ctr9 predominantly associates with active enhancers (Fig 2C, 2D, Dataset EV3). We have rephrased the sentence to better explain the logic in the revised manuscript (Page 10).

5. P. 12. The phrase "suggesting that a majority of SEs are transcriptional enhancers" is confusing. Why are they called super enhancers if they are not transcriptional enhancers? The sentence needs to be written more clearly.

Response: We have rephrased this sentence in the revised manuscript, thank you.

6. Figure 6 title. Rephrase to avoid stating Ctr9 DNA-binding. The factor precipitated need not bind DNA directly.

Response: We have rephrased this title in the revised manuscript, accordingly.

Reviewer #3 (Comments to the Authors (Required)):

In the study "The Paf1 complex positively regulates enhancer activity in mouse embryonic stem cells", Ding and colleagues reported functions of Paf1C in regulating enhancer activity, in particular in super enhancers. These results suggest the potential power of Paf1c in mECS self-renewal. This study is important to enlarge our knowledge on enhancer activity. This reviewer[er] has several comments which may be addressed.

Response: We thank reviewer #3 for this positive feedback. We have addressed his/her comments as outlined below:

1. Previous studies (e.g., Hou et al., PNAS 2019, Chen et.al., Genes & Development 2009) reported the importance of Paf1C in modulating Pol II elongation. This review[er] suggests that comparison and discussion about previous studies is needed.

Response: We appreciate this comment and we have brought this aspect into revised manuscript (page 6).

2. This review[er] didn't observe characterization of super enhancer in each cell line. Could author describe the method they used in super enhancer definition? Some cases, e.g., figure 2e, figure 3c-d, are unique enhancers in the region, which look like typical enhancer.

Response: Based on the analysis of master transcription factors and mediator, Whyte and colleagues defined 231 super enhancers and 8563 typical enhancers in mouse ESCs (Hnisz et al., 2013; Whyte et al., 2013). Our analysis is based on these two categories of enhancers. We provide this information in the text (page 9).

3. Author claimed that H3K27ac plus Ctr9 binding could improve prediction of activated enhancer. This reviewer agrees with this point by some way. While as author introduce in the manuscript, transcript enhancer (also known as enhancer RNA) could be a golden standard for active enhancer. Could author consider this important signal in definition of active enhancer?

Response: We agree to the reviewer. Indeed, we highlight the importance of eRNA transcription for active enhancers prediction in our manuscript. However, sole eRNA-based enhancer prediction could be influenced by factors, including stability of eRNA, or the sensitivity of eRNA detection. Our data suggest that the binding of Paf1C marks enhancers more reliably and robustly than eRNA transcription or H3K27ac marks alone, and therefore could serve as a better predictor of active enhancers for future studies.

4. H3K27ac is the marker for both active enhancer and promoter. If the author focus on enhancer only, other histone markers, e.g., H3K4me1, p300, are needed.

Response: We appreciate this comment by the reviewer. H3K4me1 is deemed to mark active enhancers, and more prominently poised enhancers (Creighton et al., 2010). We analyzed Ctr9 binding and H3K4me1 at both super enhancers and typical enhancers. Our analysis revealed significant overlap of Ctr9 binding with H3K4me1 at super enhancers, but at typical enhancers where many enhancers are inactive, the overlap of H3K4me1 and Ctr9 are far less significant (Figure 2C, 2D). We tried to acquire p300 occupancy profiles, but the quality of the data we obtained was not sufficient to allow meaningful interpretations. Nevertheless, we feel that the provided information is sufficient to support the claims we are making in our manuscript.

References

- Creighton, M.P., Cheng, A.W., Welstead, G.G., Kooistra, T., Carey, B.W., Steine, E.J., Hanna, J., Lodato, M.A., Frampton, G.M., Sharp, P.A., Boyer, L.A., Young, R.A., Jaenisch, R., 2010. Histone H3K27ac separates active from poised enhancers and predicts developmental state. *Proc. Natl. Acad. Sci. U.S.A.* 107, 21931–21936. doi:10.1073/pnas.1016071107
- Ding, L., Paszkowski-Rogacz, M., Nitzsche, A., Slabicki, M.M., Heninger, A.-K., de Vries, I., Kittler, R., Junqueira, M., Shevchenko, A., Schulz, H., Hubner, N., Doss, M.X., Sachinidis, A., Hescheler, J., Iacone, R., Anastassiadis, K., Stewart, A.F., Pisabarro, M.T., Caldarelli, A., Poser, I., Theis, M., Buchholz, F., 2009. A Genome-Scale RNAi Screen for Oct4 Modulators Defines a Role of the Paf1 Complex for Embryonic Stem Cell Identity. *Cell Stem Cell* 4, 403–415. doi:10.1016/j.stem.2009.03.009
- Ding, L., Paszkowski-Rogacz, M., Winzi, M., Chakraborty, D., Theis, M., Singh, S., Ciotta, G., Poser, I., Roguev, A., Chu, W.K., Choudhary, C., Mann, M., Stewart, A.F., Krogan, N., Buchholz, F., 2015. Systems Analyses Reveal Shared and Diverse Attributes of Oct4 Regulation in Pluripotent Cells. *Cell Systems* 1, 141–151. doi:10.1016/j.cels.2015.08.002
- Dorigi, K.M., Swigut, T., Henriques, T., Bhanu, N.V., Scruggs, B.S., Nady, N., Still, C.D., Garcia, B.A., Adelman, K., Wysocka, J., 2017. Mll3 and Mll4 Facilitate Enhancer RNA Synthesis and

- Transcription from Promoters Independently of H3K4 Monomethylation. *Molecular Cell* 66, 568–576.e4. doi:10.1016/j.molcel.2017.04.018
- Hein, M.Y., Hubner, N.C., Poser, I., Cox, J., Nagaraj, N., Toyoda, Y., Gak, I.A., Weisswange, I., Mansfeld, J., Buchholz, F., Hyman, A.A., Mann, M., 2015. A human interactome in three quantitative dimensions organized by stoichiometries and abundances. *Cell* 163, 712–723. doi:10.1016/j.cell.2015.09.053
- Hnisz, D., Abraham, B.J., Lee, T.I., Lau, A., Saint-André, V., Sigova, A.A., Hoke, H.A., Young, R.A., 2013. Super-enhancers in the control of cell identity and disease. *Cell* 155, 934–947. doi:10.1016/j.cell.2013.09.053
- Jahn, A., Rane, G., Paszkowski-Rogacz, M., Sayols, S., Bluhm, A., Han, C.-T., Draškovič, I., Londoño-Vallejo, J.A., Kumar, A.P., Buchholz, F., Butter, F., Kappei, D., 2017. ZBTB48 is both a vertebrate telomere-binding protein and a transcriptional activator. *EMBO reports* 18, 929–946. doi:10.15252/embr.201744095
- Kappei, D., Butter, F., Benda, C., Scheibe, M., Draškovič, I., Stevense, M., Novo, C.L., Basquin, C., Araki, M., Araki, K., Krastev, D.B., Kittler, R., Jessberger, R., Londoño-Vallejo, J.A., Mann, M., Buchholz, F., 2013. HOT1 is a mammalian direct telomere repeat-binding protein contributing to telomerase recruitment. *The EMBO Journal* 32, 1681–1701. doi:10.1038/emboj.2013.105
- Li, W., Hu, Y., Oh, S., Ma, Q., Merkurjev, D., Song, X., Zhou, X., Liu, Z., Tanasa, B., He, X., Chen, A.Y., Ohgi, K., Zhang, J., Liu, W., Rosenfeld, M.G., 2015. Condensin I and II Complexes License Full Estrogen Receptor α -Dependent Enhancer Activation. *Molecular Cell* 59, 188 – 202. doi:10.1016/j.molcel.2015.06.002
- Mikhaylichenko, O., Bondarenko, V., Harnett, D., Schor, I.E., Males, M., Viales, R.R., Furlong, E.E.M., 2018. The degree of enhancer or promoter activity is reflected by the levels and directionality of eRNA transcription. *Genes & Development* 32, 42–57. doi:10.1101/gad.308619.117
- Nitzsche, A., Paszkowski-Rogacz, M., Matarese, F., Janssen-Megens, E.M., Hubner, N.C., Schulz, H., de Vries, I., Ding, L., Huebner, N., Mann, M., Stunnenberg, H.G., Buchholz, F., 2011. RAD21 Cooperates with Pluripotency Transcription Factors in the Maintenance of Embryonic Stem Cell Identity. *PLoS ONE* 6, e19470. doi:10.1371/journal.pone.0019470.g007
- Poser, I., Sarov, M., Hutchins, J.R.A., Hériché, J.-K., Toyoda, Y., Pozniakovsky, A., Weigl, D., Nitzsche, A., Hegemann, B., Bird, A.W., Pelletier, L., Kittler, R., Hua, S., Naumann, R., Augsburg, M., Sykora, M.M., Hofemeister, H., Zhang, Y., Nasmyth, K., White, K.P., Dietzel, S., Mechtler, K., Durbin, R., Stewart, A.F., Peters, J.-M., Buchholz, F., Hyman, A.A., 2008. BAC TransgeneOmics: a high-throughput method for exploration of protein function in mammals. *Nature Methods* 5, 409–415. doi:10.1038/nmeth.1199
- Whyte, W.A., Orlando, D.A., Hnisz, D., Abraham, B.J., Lin, C.Y., Kagey, M.H., Rahl, P.B., Lee, T.I., Young, R.A., 2013. Master transcription factors and mediator establish super-enhancers at key cell identity genes. *Cell* 153, 307–319. doi:10.1016/j.cell.2013.03.035

December 7, 2020

RE: Life Science Alliance Manuscript #LSA-2020-00792-TR

Prof. Frank Buchholz
Medical Systems Biology
Medical Systems Biology, UCC, Medical Faculty Carl Gustav Carus
Medical Faculty and University Hospital Carl Gustav Carus, TU Dresden
Dresden 01307
Germany

Dear Dr. Buchholz,

Thank you for submitting your revised manuscript entitled "The Paf1 complex positively regulates enhancer activity in mouse embryonic stem cells". We would be happy to publish your paper in Life Science Alliance pending final revisions necessary to meet the reviewer 2's concerns and our formatting guidelines.

Since repeating the ChIP-Seq data was not made a requirement in the first round of review, we will editorially overrule that request for publication in LSA. However, we do suggest you clarify that the ChIP-Seq data provided are from a single run - both in the manuscript text and in the figure legend. All other concerns raised by Reviewer 2 should be addressed in the revision.

Along with the Reviewer 2's points 2-6 and the points listed at the end of this email, please also attend to the following:

- please consult our Manuscript Preparation Guidelines <https://www.life-science-alliance.org/manuscript-prep> and put your manuscript sections in the correct order
- please use the [10 author names, et al.] format in your references (i.e. limit the author names to the first 10)
- please add the supplementary figure legends to the main manuscript text
- LSA allows supplementary figures, but not EV Figures; please update your callouts for the Supplementary Figures in the manuscript Fig EV1A = Fig S1A)
- please add a callout in your main manuscript text for Figure S1B
- please add a scale bar for Figure S1B
- please rename the datasets as supplementary tables - both in their titles and in their callouts in the manuscript text
- please rename the 'Experimental Procedures' section as 'Materials and Methods'
- please provide source data (original unprocessed gels) for Figure S3B

A. FINAL FILES:

B. MANUSCRIPT ORGANIZATION AND FORMATTING:

Thank you for this interesting contribution, we look forward to publishing your paper in Life Science

Alliance.

Sincerely,

Shachi Bhatt, Ph.D.
Executive Editor
Life Science Alliance
<https://www.lsjournal.org/>
Tweet @SciBhatt @LSAJournal

Reviewer #2 (Comments to the Authors (Required)):

In this revised manuscript, Ding et al. have addressed many of the previous reviewer comments through corrections to the writing and through the presentation of additional data and figures. The results generally support the conclusions that PAF complex subunit Ctr9 occupies active super enhancers in mESCs, Ctr9 occupancy correlates with Pol II CTD Ser2P and eRNA enrichment at super enhancers, and depletion of Ctr9 mitigates the activity of super enhancers that are occupied by Ctr9 in mESCs but not the activity of super enhancers unoccupied by Ctr9. The authors conclude that Ctr9 marks active enhancers and is important for the functions of these enhancers likely by driving eRNA transcription and/or CTD Ser2 phosphorylation. They argue that Ctr9 occupancy can be used to predict active enhancers, along with eRNAs and enhancer-associated histone modifications. While the paper is improved and the conclusions are important, some issues remain.

1. The most significant issue is the lack of repetitions for the ChIP-seq data. With only a single replicate, the authors are unable to confirm the reproducibility of their data. Despite the authors' prior experience with the antibodies used, presentation of a single replicate for any experiment, in particular a mainline approach, should not be standard practice.
2. The H3K27ac and H3K4me1 enrichment profiles in the new Figure 3A heatmaps are difficult to interpret. As shown, the signals are spread out over a 2kb region and not obviously overlapping with the Ctr9 localization pattern, as would be expected from the browser tracks shown in 2A and 2F. The authors should consider adding a heatmap that spans a wider region to convince the reader that these are indeed enhancer-localized modifications and not a general background signal from their ChIP-seq experiment. While the H3K27ac signals in the browser tracks (see 2A and 2F) indicate enhancer-specific localization, this is not clear for the H3K4me1 tracks. Also, the Figure would be more informative and the authors' conclusions could be stronger if the SEs classified as Ctr9-unoccupied were also shown.
3. The authors provide percentages for SEs occupied by Ctr9, H3K4me1, H3K27ac, Ser5P and Ser2P. Presumably a cutoff was determined in classifying SEs as occupied or unoccupied. The authors should provide a description of this analysis in the Methods.
4. Page 13. "Consistent with previous reports, our analysis revealed that RNAPII Ser5p is mainly detected at the TSS regions, whereas RNAPII Ser2p is primarily found at gene bodies and exhibited maximum occupancy around transcription termination sites (TTSs) of protein coding genes, reflecting the Ctr9 occupancy pattern (Fig 3A, Fig EV4) (Grosso et al, 2012)." The authors reference to Figure 3A here is confusing. Figure 3A shows enhancers and the data are anchored relative to

the NELFa localization pattern not to the TSS.

5. Figure 1 legend. Remove "TSS of".

6. Figure 2A legend. Delete "H3K4me3". This modification is not shown.

Reviewer #3 (Comments to the Authors (Required)):

Authors addressed most of my comments. This reviewer has one minor suggestion regard my previous comment 4. It will be better if authors should clarify these details about histone modification markers, TF binding sites in revised manuscript.

Rebuttal

Reviewer #2 (Comments to the Authors (Required)):

In this revised manuscript, Ding et al. have addressed many of the previous reviewer comments through corrections to the writing and through the presentation of additional data and figures. The results generally support the conclusions that PAF complex subunit Ctr9 occupies active super enhancers in mESCs, Ctr9 occupancy correlates with Pol II CTD Ser2P and eRNA enrichment at super enhancers, and depletion of Ctr9 mitigates the activity of super enhancers that are occupied by Ctr9 in mESCs but not the activity of super enhancers unoccupied by Ctr9. The authors conclude that Ctr9 marks active enhancers and is important for the functions of these enhancers likely by driving eRNA transcription and/or CTD Ser2 phosphorylation. They argue that Ctr9 occupancy can be used to predict active enhancers, along with eRNAs and enhancer-associated histone modifications. While the paper is improved and the conclusions are important, some issues remain.

1. The most significant issue is the lack of repetitions for the ChIP-seq data. With only a single replicate, the authors are unable to confirm the reproducibility of their data. Despite the authors' prior experience with the antibodies used, presentation of a single replicate for any experiment, in particular a mainline approach, should not be standard practice.

Reply: We clarify in the text and in the figure legends that the ChIP-Seq data provided are from single runs.

2. The H3K27ac and H3K4me1 enrichment profiles in the new Figure 3A heatmaps are difficult to interpret. As shown, the signals are spread out over a 2kb region and not obviously overlapping with the Ctr9 localization pattern, as would be expected from the browser tracks shown in 2A and 2F. The authors should consider adding a heatmap that spans a wider region to convince the reader that these are indeed enhancer-localized modifications and not a general background signal from their ChIP-seq experiment. While the H3K27ac signals in the browser tracks (see 2A and 2F) indicate enhancer-specific localization, this is not clear for the H3K4me1 tracks. Also, the Figure would be more informative and the authors' conclusions could be stronger if the SEs classified as Ctr9-unoccupied were also shown.

Reply: We thank the reviewer for this suggestion. We have added heatmaps that span a wider region (50 kb) of Ctr9-bound and Ctr9-unoccupied enhancers. This data is now presented in Supplementary Figure S5.

3. The authors provide percentages for SEs occupied by Ctr9, H3K4me1, H3K27ac, Ser5P and Ser2P. Presumably a cutoff was determined in classifying SEs as occupied or unoccupied. The authors should provide a description of this analysis in the Methods.

Reply: We have added the sentence: "Enhancer occupancy by Ctr9, H3K4me1, H3K27ac, Ser5P and Ser2P was determined by detection of an at least 1 bp intersection between genomic coordinates of an enhancer region and a respective ChIP-Seq peak." in the ChIPseq Methods section to clarify this point.

4. Page 13. "Consistent with previous reports, our analysis revealed that RNAPII Ser5p is mainly detected at the TSS regions, whereas RNAPII Ser2p is primarily found at gene bodies and exhibited maximum occupancy around transcription termination sites (TTSS) of protein coding genes, reflecting the Ctr9 occupancy pattern (Fig 3A, Fig EV4) (Grosso et al, 2012)." The authors reference to Figure 3A here is confusing. Figure 3A shows enhancers and the data are anchored relative to the NELF α localization pattern not to the TSS.

Reply: The thank the reviewer to point this out. We have removed the reference to Figure 3A.

5. Figure 1 legend. Remove "TSS of".

Reply: "TSS of" was removed from the figure legend. Thank you.

6. Figure 2A legend. Delete "H3K4me3". This modification is not shown.

Reply: Change made as suggested. Thank you.

December 18, 2020

RE: Life Science Alliance Manuscript #LSA-2020-00792-TRR

Prof. Frank Buchholz
Medical Systems Biology
Medical Systems Biology, UCC, Medical Faculty Carl Gustav Carus
Medical Faculty and University Hospital Carl Gustav Carus, TU Dresden
Dresden 01307
Germany

Dear Dr. Buchholz,

Thank you for submitting your Research Article entitled "The Paf1 complex positively regulates enhancer activity in mouse embryonic stem cells". It is a pleasure to let you know that your manuscript is now accepted for publication in Life Science Alliance. Congratulations on this interesting work.

*** The manuscript text is still missing the callout for Figure S1B, and we would like to request you to add it in at the proofs stage now, to prevent any further delays in getting the paper published.***

DISTRIBUTION OF MATERIALS:

Again, congratulations on a very nice paper. I hope you found the review process to be constructive and are pleased with how the manuscript was handled editorially. We look forward to future exciting submissions from your lab.

Sincerely,

Shachi Bhatt, Ph.D.

Executive Editor

Life Science Alliance

<https://www.lsjournal.org/>

Tweet @SciBhatt @LSAjournal